# LORE: Jointly Learning The Intrinsic Dimensionality and Relative Similarity Structure from Ordinal Data

**Vivek Anand**[*]**, Alec Helbling, Mark A. Davenport, Sankaraleengam Alagapan, Christopher John Rozell**
Georgia Institute of Technology
Atlanta, GA, USA
`{vivekanand}@gatech.edu`

**Gordon J. Berman**
Emory University
Atlanta, GA, USA

## Abstract

Learning the intrinsic dimensionality of subjective perceptual spaces such as taste, smell, or aesthetics from ordinal data is a challenging problem. We introduce LORE (Low Rank Ordinal Embedding), a scalable framework that jointly learns both the intrinsic dimensionality and an ordinal embedding from noisy triplet comparisons of the form, "Is A more similar to B than C?". Unlike existing methods that require the embedding dimension to be set apriori, LORE regularizes the solution using the nonconvex Schatten-$p$ quasi norm, enabling automatic joint recovery of both the ordinal embedding and its dimensionality. We optimize this joint objective via an iteratively reweighted algorithm and establish convergence guarantees. Extensive experiments on synthetic datasets, simulated perceptual spaces, and real world crowdsourced ordinal judgements show that LORE learns compact, interpretable and highly accurate low dimensional embeddings that recover the latent geometry of subjective percepts. By simultaneously inferring both the intrinsic dimensionality and ordinal embeddings, LORE enables more interpretable and data efficient perceptual modeling in psychophysics and opens new directions for scalable discovery of low dimensional structure from ordinal data in machine learning.

## 1 Introduction

Learning subjective percepts (SPs), such as taste, smell, or aesthetic preference, poses unique challenges for machine learning. Traditional approaches rely on absolute queries that presuppose known perceptual axes. For example, a taste study might ask participants to rate stimuli on a 1-5 Likert scale (Likert, 1932) for "sweetness" or "bitterness". Such methods suffer from two critical flaws: (1) inconsistency, as respondents interpret scales differently (e.g., one person's "moderately sweet" is another's "very sweet") (Stewart et al., 2005), and (2) predefined conceptual frameworks that limit discovery by forcing ratings on predefined axes. Consequently, researchers risk missing latent dimensions (e.g., a "metallic" undertone in coffee) that participants lack vocabulary to describe.

In contrast, relative queries circumvent these issues by capturing perceptual relationships directly. For example, a triplet comparison like "Is coffee A more similar to coffee B or coffee C in taste?" allows participants to express nuanced judgments without relying on language or preset scales. Such relative comparisons are therefore particularly well suited for discovering the latent dimensions that organize subjective perceptual spaces.

Relative Similarity or Ordinal Embedding methods (OE) leverage these relative judgements to learn a multidimensional representation. However, all existing OE approaches require the user to specify

---

[*]Corresponding Author

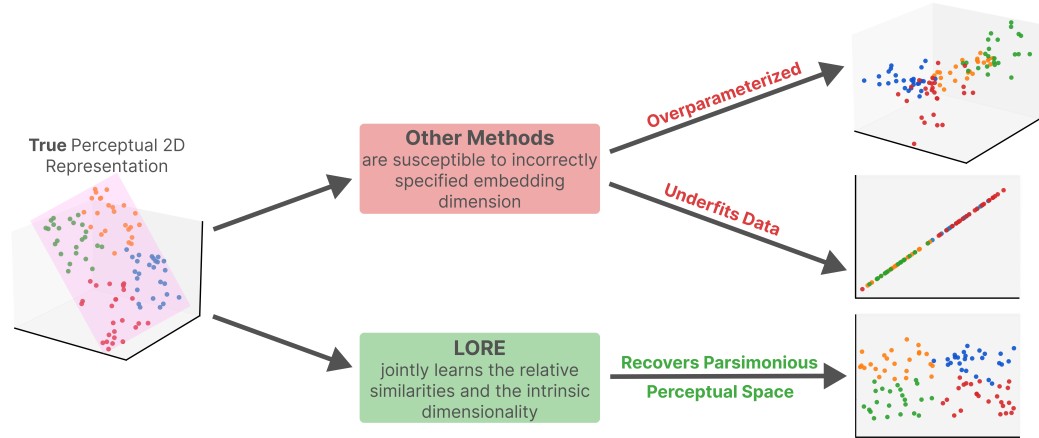

Figure 1: **LORE jointly learns both the intrinsic dimensionality and relative similarities by balancing dimensionality with similarity constraints**: Other methods are susceptible to an incorrectly chosen embedding dimension.

the embedding dimension in advance (Terada & Luxburg, 2014; Agarwal et al., 2007; Tamuz et al., 2011; Jain et al., 2016; Van Der Maaten & Weinberger, 2012), with little guidance to the "true" complexity of the perceptual space. In practice, this can lead to unnecessarily high dimensional embeddings, concealing the actual structure. For instance, an OE may perfectly satisfy all triplet constraints in a 10-dimensional space, even if the underlying percept is only 2-dimensional.

Scientific discovery demands parsimony, a principle formalized as Occam's razor (Bishop & Nasrabadi, 2006). For the taste example, a 2D embedding is preferable to a 10D alternative: it is easier to interpret, less computationally intensive, and more useful for downstream analyses. In practical terms, a 10D taste embedding might fragment "sweetness" into several axes, complicating flavor design or neurological interpretation. Yet, most OE approaches, despite high triplet accuracy, produce overly complex models that mask the true structure of the latent percept.

To address this gap, We introduce LORE, a new ordinal embedding algorithm that jointly learns both the embedding and the intrinsic dimensionality, instead of needing to specify the dimension apriori. LORE regularizes using the nonconvex Schatten-$p$ quasi-norm, explicitly balancing triplet accuracy with representation compactness and is optimized via an iteratively reweighted algorithm, with guarantees of convergence to stationary points. Our main contributions are:

1. **LORE, a novel ordinal embedding algorithm that recovers latent representations that match the intrinsic dimensionality of human perceptual similarity data.** LORE jointly infers both the embedding and its dimensionality by regularizing with the nonconvex Schatten-$p$ quasi-norm. By balancing triplet accuracy and rank regularization we can infer a compact yet accurate representation avoiding underfitting or overparameterization. We optimize the resulting objective using an iteratively reweighted schatten quasi norm algorithm, and provide convergence guarantees of the OE to stationary points.

2. **LORE reliably uncovers the intrinsic dimensionality of data through an extensive evaluation where the dimensionality is known apriori.** We first extensively test our algorithm on data with various dimensionality, noise levels and number of queries and demonstrate it outperforms existing methods by far in estimating intrinsic dimensionality with close to optimal performance in triplet accuracy. Secondly, we conduct a simulated perceptual experiment to model taste using an LLM as the ground truth perceptual space we try to model. See Figure 1 for a high level summary of our results.

3. **LORE outperforms numerous state of the art methods on the large crowd sourced datasets and learns semantically interpretable axes.** We find that LORE achieves a lower rank representation compared to all baselines while achieving comparable triplet accuracy on three separate crowdsourced datasets (Wilber et al., 2014; Kleindessner & Von Luxburg, 2017; Ellis et al., 2002) and learns axes which are semantically interpretable.

We anticipate LORE will be a valuable tool for mapping subtle subjective phenomena to interpretable low-dimensional spaces across psychology, neuroscience, and social science. By removing the need to hand tune embedding dimension, LORE jointly learns both the relative similarities and recover true intrinsic dimensionality enabling data driven discovery of subjective percepts.

## 2 RELATED WORK

**Ordinal Embeddings as Tools for Psychophysical Scaling:** Psychophysics aims to discover the quantitative mappings that humans use to connect external stimuli to inner perceptual experiences. Psychological percepts are usually studied via *relative judgements* as they are less prone to individual biases, scale interpretation and memory limitations than absolute judgements as humans do not perceive stimuli in isolation (Stewart et al., 2005). Given the constraints of data collection, a core challenge in psychophysics is reconstructing perceptual spaces from a few human similarity judgments. OEs address this challenge; they are both query efficient and capable of reconstructing multidimensional perceptual spaces. For example, Filip et al. (2024) derived a tactile-visual embedding for wood textures, identifying roughness and gloss as perceptually orthogonal dimensions. Huber et al. (2024) used one such OE to map philosophical concepts onto conceptual axes from human similarity data and Sauer et al. (2024) used it to map perceived distortions of vision from spectacles. Moreover, active learning approaches like Canal et al. (2020) have demonstrated how query efficiency for data collection can be further improved.

**Metric Learning/Contrastive Learning are distinct from OEs:** Learning from relative comparisons has been used in metric learning (Suárez-Díaz et al., 2018) and contrastive learning (Chen et al., 2020). Metric learning aims to learn a metric space from the data while contrastive learning separates similar and dissimilar datapoints. Metric Learning typically combines relative judgements and explicit representations (say images). The goal is to learn a distance metric from both sources of information. This additional representation, absent in OEs, changes the optimization problem and prevents direct transfer of metric learning approaches. Contrastive learning seeks to group similar datapoints together and push dissimilar ones apart with the presence of explicit additional supervised information which do not exist for OEs. Therefore, while metric learning and contrastive learning methods are similar in learning from relative information, they cannot be directly applied to OEs.

**Intrinsic Dimensionality recovery is critical for psychophysics:** A core goal in psychophysics is to recover the latent internal representations that individuals use to perceive psychophysical stimuli. Each representation is composed of two important characteristics: how well the representation recovers the ordinal relationships between the percepts and the *intrinsic rank or dimensionality* of the representation obtained. While OEs are able to maintain ordinal consistency (Vankadara et al., 2023), they are unable to identify the intrinsic dimensionality as we show in this paper. This is a key limitation of OEs that reduces their utility for psychophysical analysis. Künstle et al. (2022) addressed this problem by modelling it as a multiple hypothesis test with separate embeddings trained for each candidate dimension and triplet accuracies used to estimate the true intrinsic rank. This approach, however, has two main limitations:

1. **Hypothesis dependence**: It requires predefining plausible dimensionalities, risking model misspecification and reducing statistical power if the true dimensionality exceeds or is less than the hypothesized bounds.

2. **Lack of Scalability**: Training multiple embeddings for each hypothesized rank is computationally expensive and quickly becomes prohibitive for a greater number of percepts. This is especially problematic for active querying where efficiency is critically important.

Building on these limitations, we propose a method to **jointly infer both dimensionality and multidimensional representations** via a novel OE method, eliminating the need for explicit hypothesis enumeration. For psychophysics, this enables recovery of perceptual geometry without prior assumptions on dimensionality. For machine learning, it offers a scalable approach to uncovering low dimensional structure directly from ordinal data.

Table 1: Characterization of Different Ordinal Embedding Algorithms

| Method | Optimizes Over | Recovers Intrinsic Rank | Scalable | High Triplet Accuracy | Semantically Interpretable |
|---|---|---|---|---|---|
| GNMDS | Gram Matrix | ✗ | ✗ | ✗ | ✗ |
| CKL | Gram Matrix | ✗ | ✗ | ✓ | ✓ |
| FORTE | Gram Matrix | ✗ | ✓ | ✓ | ✗ |
| t-STE | Embedding | ✗ | − | ✓ | ✗ |
| SOE | Embedding | ✗ | ✓ | ✓ | ✗ |
| OENN | Embedding | ✗ | ✓ | ✗ | ✗ |
| **LORE** (ours) | Embedding | ✓ | ✓ | ✓ | ✓ |

## 3 BACKGROUND ON ORDINAL EMBEDDINGS

The ordinal embedding problem seeks to learn an embedding matrix $\mathbf{Z} \in \mathbb{R}^{N \times d'}$ from triplet judgements from the true perceptual space lying in an unknown $\mathbf{P} \in \mathbb{R}^{N \times d}$ where $d \ll N$ is the *intrinsic dimensionality* or the *intrinsic rank* of the perceptual space. OE problems usually assume integral dimensionalities/ranks due to low number of percepts and we do the same. $\mathbf{Z}$ is learned indirectly via noisy *similarity triplet* comparisons where the anchor percept $a$ is more similar or closer in the perceptual space to percept $i$ than percept $j$ into an embedding space of dimension $d'$. Specifically, this is denoted by $(a, i, j) = t \in T$ where $d(\mathbf{P}_{a,:}, \mathbf{P}_{i,:}) < d(\mathbf{P}_{a,:}, \mathbf{P}_{j,:})$ where $\mathbf{P}_{a,:}, \mathbf{P}_{i,:}, \mathbf{P}_{j,:}$ are the rows indexed by percepts $a, i, j$ respectively in $\mathbf{P}$ and $d(\cdot, \cdot)$ is the Euclidean distance between the unknown percepts. A central challenge is that intrinsic rank is unknown and the embedding dimension is set heuristically.

Though this framework is relatively simple, solving OEs efficiently can be challenging as the OE problem is NP-Hard (Bower et al., 2018), most loss functions are nonconvex and efficient learning demands at least $\mathcal{O}(Nd \log N)$ actively sampled triplets (Jain et al., 2016). As a result, the choice of optimization framework is crucial to obtaining a good OE and depending on dataset characteristics, different OE methods may be preferred for different situations (Vankadara et al., 2023).

Gram matrix approaches optimize a positive semidefinite matrix $\mathbf{G} = \mathbf{Z}\mathbf{Z}^T \in \mathbb{R}^{N \times N}$ that capture the pairwise differences. While theoretically appealing because they are agnostic to the embedding dimension during optimization, they require enforcing PSD constraints that are not scalable for large $N$. Early methods like Generalized Non Metric Multi Dimensional Scaling (GNMDS) (Agarwal et al., 2007) and probabilistic models like Crowd Kernel Learning (CKL) suffer from limited accuracy or poor scalability. Fast Ordinal Triplet Embedding (FORTE) accelerates this with a kernelized nonconvex triplet loss optimized by efficient Projected Gradient Descent (PGD) and line search.

Direct embedding approaches optimize $\mathbf{Z}$ which leads to faster gradient updates that scale with the smaller $\mathcal{O}(Nd')$ versus $\mathcal{O}(N^2)$ with Gram matrix approaches. Examples include t-distributed Stochastic Triplet Embedding (t-STE) (Van Der Maaten & Weinberger, 2012) and Soft Ordinal Embedding (SOE) (Terada & Luxburg, 2014) with the latter widely used for its efficiency and high accuracy. A deep learning variant, Ordinal Embedding Neural Network (OENN) (Vankadara et al., 2023) underperforms likely due to the limited supervisory signal in purely ordinal data.

However, a shared fundamental limitation of all existing methods is the inability to recover the intrinsic rank $d$, which risks overparameterizing the true perceptual latent space.

## 4 METHODS

We introduce a scalable ordinal embedding (OE) framework that jointly learns both the embedding and the intrinsic rank of the perceptual space. To ensure computational efficiency on large datasets (large $T$ and $N$) we directly optimize the embedding $\mathbf{Z}$ instead of the Gram matrix $\mathbf{G}$. The key insight is that we want the learning algorithm to adaptively select the embedding dimensionality as needed to fit the percepts well but not use any more extra space than necessary. Therefore, a natural approach is to penalize the rank of the learned embedding via regularization.

As SOE has the best properties of all the OEs that optimize over $\mathbf{Z}$, we extend it with regularization. As the rank constraint is NP-Hard and non-convex (Fazel et al., 2001) a common approach is to regularize with the nuclear norm instead where $\|\mathbf{Z}\|_* = \sum_{i=1}^{\min\{N,d'\}} \sigma_i(\mathbf{Z})$, where $\sigma_i(\mathbf{Z})$ is the $i$-th singular value (Candes & Recht, 2008; Fazel et al., 2001). The objective then becomes:

$$\min_{\mathbf{Z}} \quad \Psi(\mathbf{Z}) = \sum_{(a,i,j)\in T} \max\{0, 1 + d(\mathbf{Z}_{a,:}, \mathbf{Z}_{i,:}) - d(\mathbf{Z}_{a,:}, \mathbf{Z}_{j,:})\} + \lambda\|\mathbf{Z}\|_*. \tag{1}$$

Though the nuclear norm is convex and relatively easy to optimize, it uniformly shrinks all of singular values (Negahban & Wainwright, 2011; Zhang, 2010). Recent theoretical and empirical evidence indicates that the nonconvex Schatten-$p$ quasi norm $\|\mathbf{Z}\|_p^p = \sum_{i=1}^{\min\{N,d\}} \sigma_i(\mathbf{Z})^p = \sum_{i=1}^{\min\{N,d\}} g[\sigma_i(\mathbf{Z})]$ for $0 < p < 1$, recovers the intrinsic rank for low rank recovery problems better than the nuclear norm can (Lu et al., 2014; Marjanovic & Solo, 2012). The Schatten-$p$ quasi norm generalizes the nuclear norm by penalizing larger singular values less severely which is shown to aid in intrinsic rank recovery. We leverage this property and for the first time, to our knowledge, integrate the Schatten quasi-norm into a scalable ordinal embedding framework, allowing implicit perceptual rank discovery as seen below in:

$$\min_{\mathbf{Z}} \quad \Psi(\mathbf{Z}) = \sum_{(a,i,j)\in T} \max\{0, 1 + d(\mathbf{Z}_{a,:}, \mathbf{Z}_{i,:}) - d(\mathbf{Z}_{a,:}, \mathbf{Z}_{j,:})\} + \lambda\|\mathbf{Z}\|_p^p. \tag{2}$$

Though incorporating the Schatten Quasi-Norm improves rank recovery properties, it also introduces additional nonconvexity into the regularizer that makes optimization more challenging. To overcome the inherent non-differentiability of the ordinal loss and the complexity of nonconvex regularization, we smooth the hinge triplet loss with the softplus function (Dugas et al., 2001). This transformation makes the objective differentiable except where the embedding collapses ($\mathbf{Z}_{a,:} = \mathbf{Z}_{i,:}$ or $\mathbf{Z}_{a,:} = \mathbf{Z}_{j,:}$). However, collapses can be avoided with wide initializations of $\mathbf{Z}$. This smoothing enables provable convergence and is empirically essential, as it mitigates zero gradient plateaus to facilitate training on large datasets. Then the objective function is defined as:

$$\min_{\mathbf{Z}} \quad \Psi(\mathbf{Z}) = \sum_{(a,i,j)\in T} \log(1 + \exp(1 + d(\mathbf{Z}_{a,:}, \mathbf{Z}_{i,:}) - d(\mathbf{Z}_{a,:}, \mathbf{Z}_{j,:}))) + \lambda \sum_{i=1}^{\min\{N,d'\}} \sigma_i(\mathbf{Z})^p. \tag{3}$$

Despite smoothing the ordinal loss, our objective remains highly nonconvex due to the Schatten-$p$ quasi-norm regularization, which makes reliable optimization difficult. Standard gradient methods often get stuck in poor local minima or fail to converge. To overcome this, we use an iteratively reweighted algorithm inspired by Sun et al. (2017). At each step, the algorithm minimizes a weighted surrogate of the original objective, leading to steady improvement even in complex landscapes. As established in Theorem 1, this procedure is guaranteed to converge to a stationary point, ensuring robust and reliable learning.

**Theorem** (LORE converges to a stationary point). *The sequence of OEs generated by the LORE algorithm* $\{\mathbf{Z}^k\}_{k=1,2,3,...}$ *converges. i.e.*

$$\sum_{k=1}^{+\infty} \|\mathbf{Z}^{k+1} - \mathbf{Z}^k\|_F < +\infty \tag{4}$$

*Proof Sketch:* We use the general framework for nonconvex Schatten Quasi-Norm optimization as seen in Sun et al. (2017) but crucially, check the specific conditions for the LORE objective. The full proof is in Appendix A.

Our convergence guarantee is significant because, for ordinal embedding problems, stationary points are widely believed to be nearly as good as global optima in objective value. This is supported empirically (Vankadara et al., 2023) and theoretically. Bower et al. (2018) proved that for certain OE settings with $d = 2$, all local optima are global. Moreover, when sufficient triplet data is available, sub-optimal local minima are rarely observed. Building on these insights, we expect that our method will also recover high quality embeddings in realistic settings. Our experimental results

---

**Algorithm 1** Learning LORE

---

1: **Input:** $\mathbf{Z}^0 \in \mathbb{R}^{N \times d'}, T, \lambda$
2: Assign prev_objs $\leftarrow [\infty]$
3: **for** $k = 0, 1, 2, \ldots$ **do**
4:      $\sigma \leftarrow$ Singular Values($\mathbf{Z}^k$)
5:      curr_obj $\leftarrow \sum_{(a,i,j) \in T} \log(1 + \exp(1 + d(\mathbf{z}_a, \mathbf{z}_i) - d(\mathbf{z}_a, \mathbf{z}_j))) + \lambda \sum_{i=1}^{\min\{N, d'\}} \sigma_i(\mathbf{Z})^p$
6:      **if** $|\text{curr\_obj} - \text{prev\_objs}[-1]| < \text{tol}$ **then**
7:          **break**                                                 {Convergence check}
8:      **end if**
9:      $\mathbf{U}, \mathbf{S}, \mathbf{V}^T \leftarrow \text{SVD}(\mathbf{Z}^k - \frac{1}{\mu} \nabla_{\mathbf{Z}^k} f(\mathbf{Z}^k))$
10:     $\mathbf{S}^k \leftarrow \mathbf{S} - \frac{p}{\mu} \sigma^{p-1}$
11:     $\mathbf{S}^k \leftarrow \text{sorted}(\mathbf{S}^k[\mathbf{S}^k > 0], \text{descending})$
12:     $\mathbf{Z}^{k+1} \leftarrow \mathbf{U}\mathbf{S}^k\mathbf{V}^T$
13:     prev_objs$[k] \leftarrow$ curr_obj
14:     **if** $\|\mathbf{Z}^{k+1} - \mathbf{Z}^0\|_\infty < \text{tol}$ **then**
15:        **break**                                      {Check if close to stationary point}
16:     **end if**
17: **end for**
18: **return** $\mathbf{Z}^{k+1}$

---

confirm that LORE learns high accuracy ordinal embeddings, even with the inherent nonconvexity of the objective.

We implement the optimization using an efficient iteratively reweighted algorithm, seen in Algorithm 1, that updates the embedding and regularization at each step. In the typical regime where the embedding dimension is much smaller than the number of items and triplets, each iteration requires $\mathcal{O}(d'(T + Nd'))$ operations, making LORE scalable to large datasets. Additional implementation specifics are in Appendix B.

In summary, our methodological contributions are: (1) formulating a new ordinal embedding approach that jointly learns the ordinal embedding and intrinsic rank using Schatten quasi-norm regularization; (2) establishing an efficient optimization strategy based on iteratively reweighted minimization tailored for this nonconvex objective, along with convergence guarantees and (3) providing a scalable algorithm suitable for large scale perceptual similarity data.

## 5 RESULTS

We present five pieces of empirical evidence in support of our method's claims. We outline our experimental setup, including baseline methods and the generative process for producing ordinal embedding (OE) tasks. Next, we show how to select regularization levels for LORE. We then benchmark LORE against standard baselines across key metrics, followed by a comparison on proxy large language model (LLM) generated perceptual spaces. Finally, we assess performance on real, crowdsourced triplet data involving human judgments and see that LORE's learned axes have semantic meaning. Collectively, these results demonstrate that LORE is the only OE method that can effectively jointly learn high quality ordinal embeddings and the intrinsic rank.

### 5.1 SETUP

We benchmark LORE primarily against (1) SOE and (2) FORTE. These methods represent the best performing direct and Gram matrix OE approaches, respectively, as established by prior work (Vankadara et al., 2023). For our last two experiments, which are computationally less demanding, we also compare against additional OE methods like t-STE, CKL and the dimensionality estimation method using hypothesis testing from (Künstle et al., 2022) denoted as Dim-CV Our core evaluation criteria are:

- **Test Triplet Accuracy**: The proportion of held-out triplets correctly satisfied by the learned embedding and the primary metric in the OE literature.

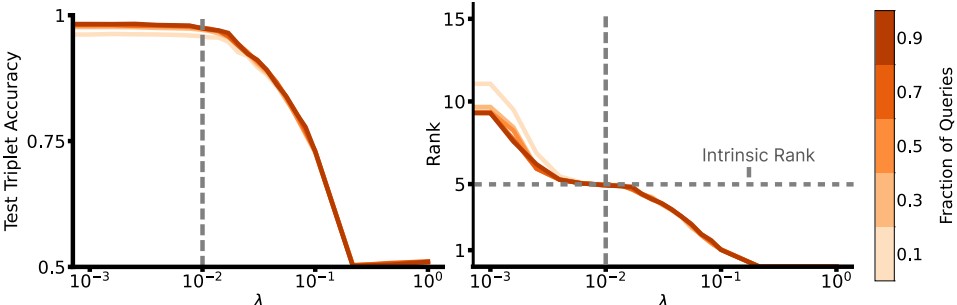

Figure 2: **LORE has high test triplet accuracy and intrinsic rank recovery across varying number of queries**. (Left) Mean test triplet accuracy vs $\lambda$ for LORE as Fraction of Queries varies. (Right) Mean measured rank vs $\lambda$ for LORE as Fraction of Queries varies.

- **Measured Rank**: The effective rank of the learned embedding, as a measure of intrinsic dimensionality recovery.

An ideal ordinal embedding should achieve high test triplet accuracy while maintaining a measured rank close to the true intrinsic rank of the underlying perceptual space.

For LORE, we fix $p = 0.5$ as it offers a good balance between rank recovery and optimization stability as seen in prior work (Lu et al., 2014; Wang et al., 2024; Sun et al., 2017). Our Results in Appendix G concur with the prior work. As $\mu$ only needs to be greater than the Lipschitz constant of triplet loss term, we set it to $0.1$ which is greater than what we find empirically. Both of these are set apriori and do not require tuning. The only hyperparameter that is tuned is the regularization parameter $\lambda$. However, as we show in Section 5.2, there exists a wide range of $\lambda$ around $0.01$ that yields both high triplet accuracy and intrinsic rank recovery across a variety of dataset conditions. Thus, we expect that in practice, a user can set $\lambda$ to $0.01$ without extensive hyperparameter tuning which our later experiments confirm works well across real world datasets. Initializations are usually critical for nonconvex optimization problems. However, due to the guaranteed convergence of LORE to stationary points, and the fact that Ordinal Embedding algorithms tend to have good local minima we find that random Gaussian initializations with variance of at least 5 work well across all experiments without any special tuning. All of our experiments use this initialization scheme and we suggest it as a default choice for practitioners.

We systematically vary four factors: fraction of queries, intrinsic rank, number of percepts, and noise level in the generative model. For synthetic experiments, we generate perceptual spaces of specified size and rank, sample noisy triplets to mimic human responses, and fit each method before evaluating on test triplets. Our synthetic data model generates a random perceptual space of specified rank and number of percepts, followed by sampling triplets with replacement to simulate human queries. We then use a standard approach (Canal et al., 2020; Vankadara et al., 2023) to model response uncertainty by sampling Gaussian noise independently and adding to each triplet distance. The resulting triplet data is then used to fit all OE algorithms, which are evaluated on held-out test triplets for both accuracy and measured rank. Unless otherwise stated, all experiments use query_fraction $= 0.1$, $p = 0.5$, $d = 5$, $N = 50$, noise $= 0.1$, and $d' = 15$ for 30 independent seeds. We include code to reproduce this generative process in the supplemental material.

## 5.2 LORE HAS A CONSISTENT AND STABLE REGULARIZATION SETTING THAT YIELDS HIGH TRIPLET ACCURACY AND INTRINSIC RANK RECOVERY

Our first set of experiments explores whether LORE admits a regularization regime that yields both high test triplet accuracy and reliable intrinsic rank recovery. As shown in Figure 2, across a broad range of the regularization parameter ($\lambda \approx 0.01$), LORE achieves nearly perfect test triplet accuracy and accurate intrinsic rank recovery, even as the fraction of queried triplets varies. Further results in Appendix C show that LORE performs similarly with varying noise, number of percepts and intrinsic rank. These pieces of evidence taken together confirm that high triplet accuracy and intrinsic rank recovery persists with different noise levels, numbers of percepts and intrinsic rank for the same regularization range. Thus, LORE is robust in hyperparameter selection ($\lambda$).

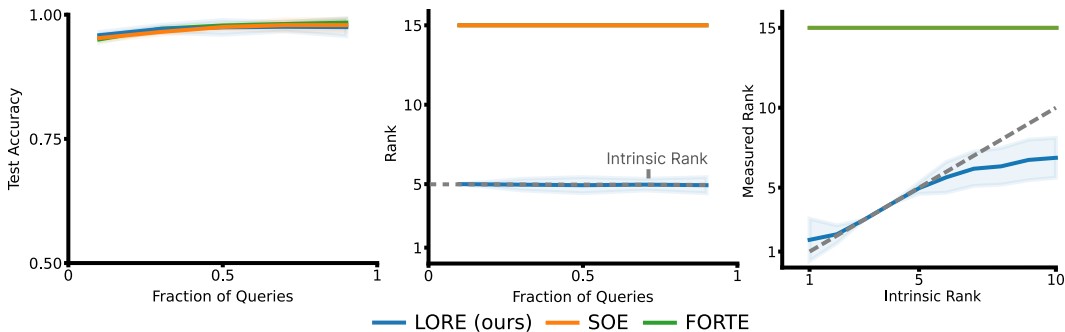

Figure 3: **Only LORE can recover the intrinsic rank while maintaining comparable test triplet accuracy as number of queries varies**: (Left) Mean test triplet accuracy vs fraction of queries used. (Center) Mean measured rank vs fraction of queries used. (Right) Mean Measured Rank vs Intrinsic Rank. The gray dotted line indicates the ideal case where the measured rank is equal to the intrinsic rank. Shaded Areas indicate $\pm 2$ Standard Deviations.

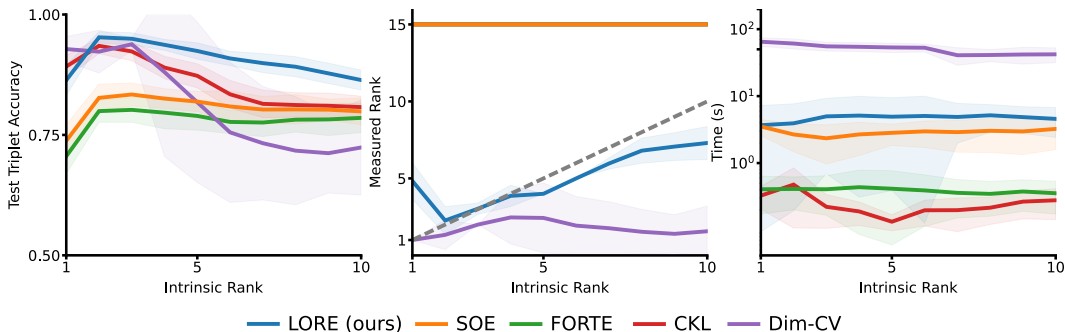

Figure 4: **LORE outperforms baselines for both test triplet accuracy and intrinsic rank for a simulated LLM perceptual experiment**. (Left) Mean test triplet accuracy vs intrinsic rank. (Center) Mean measured rank vs intrinsic rank. The gray dotted line is the ideal case where the measured rank is equal to the intrinsic rank. (Right) Time taken for processing vs intrinsic rank. Shaded Areas indicate $\pm 2$ Standard Deviations.

### 5.3 LORE OUTPERFORMS BASELINES IN RANK RECOVERY; MATCHES IN TRIPLET ACCURACY

Figure 3 (left, center) demonstrates that LORE uniquely recovers the true intrinsic rank of the embedding across all tested query fractions, while baseline methods consistently default to the maximum allowed dimension. Importantly, LORE matches the test triplet accuracy of the best baseline across all conditions, achieving low rank solutions without sacrificing predictive performance.

Additionally, Figure 3 (right) shows that as the true intrinsic rank increases, only LORE tracks this change, whereas all other methods ignore the underlying complexity and fail to adapt. While some loss in rank recovery is observed at higher true ranks (expected due to fixed number of triplets and the curse of dimensionality (Bishop & Nasrabadi, 2006)), LORE consistently outperforms competitors in recovering reduced the intrinsic rank. Further results in Appendix D confirm that LORE maintains an advantage across different noise and percept counts. Thus, LORE is the only method to reliably recover both accurate ordinal embeddings and the intrinsic rank thereby avoiding underfitting or overparameterizing the true perceptual space. These results highlight the practical value of LORE for applications where discovering latent structure is critical.

### 5.4 LORE RECOVERS INTRINSIC RANK IN A SIMULATED PERCEPTUAL EXPERIMENT

Human perceptual experiments are a key application of LORE, but for real perceptual datasets, the true intrinsic rank is unknown. Consequently, we leverage recent findings that large language models (LLMs) encode human-aligned perceptual information across domains such as taste, pitch, and timbre (Marjieh et al., 2024). Therefore we use the LLM embedding space as a realistic proxy

of the true perceptual space in humans. To evaluate how our LORE compares to other baseline algorithms and the Dim-CV, which uses hypothesis testing and training multiple embeddings to estimate intrinsic rank, we evaluate both the test triplet accuracy and the measured rank of the learned embeddings. For Dim-CV we keep the range of hypothesed dimensions from 1 to 10 to match the true intrinsic ranks we test here. The measured rank is the averaged resulting rank of the embedding learned by each algorithm across 30 independent runs.

We first obtain SBERT embeddings (Reimers & Gurevych, 2019) for 50 randomly chosen foods, then control the intrinsic dimensionality for this experiment by applying truncated singular value decomposition (ranks 1-10). From this lower dimensional space, we generate noisy triplet comparisons (sampling 5% of the total, 30 runs, noise=0.1) to mimic the data limited regime typical in human perceptual experiments. A detailed experimental setup is included in Appendix I.

Figure 4 summarizes the results. Even in this highly undersampled setting, LORE closely tracks the intrinsic rank across all tested values, while all baseline OE algorithms default to the embedding dimension. Dim-CV not only is farther from the intrinsic rank than LORE but also takes considerably longer to run (note the log scaled y axis!) due to training multiple ordinal embeddings from hypothesis testing and cross validation. Also, LORE significantly outperforms baselines in test triplet accuracy, demonstrating robust ordinal embedding recovery even with noise, small amounts of data and realistic semantic structure in the percept while still reliably recovering the intrinsic rank.

## 5.5 LORE LEARNS LOW RANK AND ACCURATE REPRESENTATIONS ON CROWDSOURCED DATA

To test LORE in practical, noisy settings with unknown intrinsic rank, we evaluate it alongside baselines on three representative crowdsourced human similarity datasets covering food images (Wilber et al., 2014), material images (Lagunas et al., 2019), and car images (Kleindessner & Von Luxburg, 2017). These datasets differ in size, query semantics, and noise, reflecting the variety and challenges of real world ordinal data with results in Table 2. Further details are in Appendix E with additional real world dataset evaluations in Appendix K.

All methods are trained on a random sample comprising of 90% of the total triplets, Gaussian noise and the same embedding dimension. Across all datasets, LORE has comparable test triplet accuracy to existing methods, but uniquely yields a substantially lower rank (for e.g., Food-100: LORE gets rank 3.3 vs. 15 for the others), suggesting a low rank structure. For materials, LORE gets both the highest triplet accuracy and a low rank structure of ~2-3 dimensions verified in UMAP visualizations from (Lagunas et al., 2019). For cars, extreme noise is reflected in low accuracy for all methods yet LORE's embeddings are more compact and do not degenerate to random chance like Dim-CV. There is considerable variance in time taken across methods and datasets due to the optimization characteristics of each method but LORE is the second fastest method after FORTE consistently.

Dim-CV performs poorly, likely due to its conservative approach in dimensionality selection by hypothesis testing, underfitting compared to the other methods. Dim-CV is restricted to a realistic dimensionality range (1-10) for data with unknown intrinsic rank. LORE not only learns low-rank representations but also maintains competitive triplet accuracy compared to methods that overparameterize the space, without needing dimensionality assumptions. These results show that only LORE recovers low rank structure (avoiding overparameterizing) without sacrificing significant accuracy (avoiding underfitting) from real data, enabling practical perceptual modeling.

## 5.6 LORE'S LEARNED AXES ARE SEMANTICALLY INTERPRETABLE

Figure 5 shows that, without semantic supervision, LORE's first three axes for Food-100, the same embedding used for Table 2, each align with interpretable food properties: from sweet to savory (Axis 1), dense to light (Axis 2), and carb-rich to protein/vegetable (Axis 3). The last axis is slightly less coherent, as is expected for axes linked to smaller singular values. These results demonstrate that LORE actually recovers semantically meaningful latent dimensions while recovering a low rank embedding. Consequently, the axes are interpretable and this property is invaluable for scientific discovery where the subjective percept is not well understood. Additional interpretability results for the other methods are included in Appendix F show that LORE's axes are consistently more aligned with meaningful semantic concepts than every method except CKL which is comparable to it.

Table 2: Comparison of OEs on Real Life Ordinal Datasets

| Method | Food-100 | | | Materials | | | Cars | | |
|---|---|---|---|---|---|---|---|---|---|
| | Test Acc. | Rank | Time (s) | Test Acc. | Rank | Time (s) | Test Acc. | Rank | Time (s) |
| **LORE** (Ours) | 82.45 ± 0.27 | **3.3** ±**0.47** | 6.64 ± 3.90 | 84.08 ± 0.19 | **2.23** ±**0.43** | 5.77 ± 3.37 | 52.12 ± 1.22 | **3** ±**0.45** | 4.45 ± 1.62 |
| **SOE** | 82.34 ± 0.32 | 15 ± 0.00 | 27.09 ± 1.38 | 81.86 ± 0.55 | 15 ± 0.0 | 35.48 ± 1.30 | 53.17 ± 1.42 | 15.0 ± 0.0 | 5.53 ± 1.22 |
| **FORTE** | 81.73 ± 0.46 | 15 ± 0.00 | 6.34 ± 0.52 | 79.35 ± 0.77 | 15 ± 0.0 | 3.74 ± 0.58 | 52.91 ± 0.84 | 15.0 ± 0.0 | 0.85 ± 0.18 |
| **t-STE** | 82.79 ± 0.24 | 15 ± 0.00 | 40.93 ± 20.14 | 83.44 ± 0.49 | 15 ± 0.0 | 27.15 ± 3.16 | 53.70 ± 1.15 | 15.0 ± 0.0 | 15.13 ± 4.29 |
| **CKL** | 82.75 ± 0.20 | 15 ± 0.00 | 18.41 ± 7.89 | 83.94 ± 0.11 | 15 ± 0.0 | 14.77 ± 1.79 | 54.06 ± 1.19 | 15.0 ± 0.0 | 4.85 ± 0.39 |
| **Dim-CV** | 77.67 ± 0.02 | 1.47 ± 0.51 | 1721.9 ± 26.71 | 78.10 ± 3.79 | 1.0 ± 0.0 | 1428.6 ± 32.84 | 50.43 ± 1.07 | 1.0 ± 0.0 | 270.56 ± 8.86 |

Figure 5: **LORE's learned axes are semantically interpretable**: Food groups as axis value varies for the first three learned axes of the LORE embedding learned on the Food-100 dataset. (Same embedding as one learned for Table 2).

## 6 DISCUSSION

In this work, we introduced LORE, a framework for jointly learning the intrinsic rank and the true perceptual latent space via an ordinal embedding. Our results show that LORE consistently recovers low-dimensional representations, with ranks that closely match ground truth while maintaining competitive test triplet accuracy. Moreover, we show that LORE does not underfit the perceptual space like Dim-CV does yet does not overparameterize the space like all other baselines do. On real crowdsourced data, LORE also uncovers interpretable axes aligned with meaningful semantic concepts, making subjective perceptual spaces easier to analyze.

One limitation is the absence of theoretical guarantees for exact rank recovery or optimal embeddings. Our method empirically performs well, but its theoretical underpinnings remain an open question as LORE's optimization is only guaranteed to reach stationary points, not global minima.

Future directions include developing theoretical guarantees and active learning methods to collect perceptual data more efficiently. We hope our work inspires further applied and theoretical advances applying LORE to uncover the structure of perceptual spaces across a range of domains.

## REPRODUCIBILITY STATEMENT

We have taken several steps to ensure the reproducibility of our work. All code and detailed documentation for our artificial human experiments and crowdsourced human experiments are included in the supplemental material, with full experimental configurations provided in Appendix I and Appendix J.

For the synthetic experiments and baseline comparisons, which require large scale parallelization and substantial computational resources, we do not provide raw code. Instead, we describe in detail the procedures and parameter settings necessary to reproduce them in Appendix H.

Our main theoretical result, establishing convergence of LORE to a local optimum, includes a complete proof with all required assumptions in Appendix A. To support practical use, we provide a demo (in the supplemental material) showing how LORE can be applied to new datasets, along with additional implementation details in Appendix B.

The code to reproduce our results and a demo on how to use LORE are in https://github.com/vivek2000anand/lore_iclr. LORE's integration into cblearn (Künstle & von Luxburg, 2024), a Python package for ordinal embeddings and comparison-based machine learning is forthcoming. We believe these efforts will make our work fully reproducible and accessible to the community.

### ACKNOWLEDGMENTS

We would like to sincerely thank Belén Martín-Urcelay, Yenho Chen and Chiraag Kaushik for helpful discussions and feedback on the paper.

This work was supported by the National Science Foundation under Grant No. CCF-2107455. Additionally, supported in part by the NIH BRAIN Initiative through NIMH grant R61MH138966, the National Center for Advancing Translational Sciences of the National Institutes of Health under Award Number UL1TR002378 and KL2TR002381, and the Julian T. Hightower Chair at Georgia Tech. Vivek Anand was supported by the National Institutes of Health under the Data Science in Computational Neural Engineering Training Grant T32EB025816 and the National Defense Science and Engineering Graduate (NDSEG) Fellowship Program. The content is solely the responsibility of the authors and does not necessarily represent the official views of the National Institutes of Health.

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

## A  PROOF FOR THEOREM 1

**Theorem 1 (LORE converges to a stationary point).** *The sequence of OEs generated by the LORE algorithm* $\{\mathbf{Z}^k\}_{k=1,2,3,\dots}$ *converges. i.e.*

$$\sum_{k=1}^{+\infty} \|\mathbf{Z}^{k+1} - \mathbf{Z}^k\|_F < +\infty \tag{5}$$

*Proof.* We use the general framework for nonconvex Schatten Quasi-Norm optimization as seen in (Sun et al., 2017) but check the specific conditions for the LORE objective.

Let us split up the objective as follows.

$$\min_{\mathbf{Z}} \quad \Psi(\mathbf{Z}) = \sum_{(a,i,j)\in T} \log(1 + \exp(1 + d(\mathbf{Z}_{a,:}, \mathbf{Z}_{i,:}) - d(\mathbf{Z}_{a,:}, \mathbf{Z}_{j,:}))) + \lambda\|\mathbf{Z}\|_p^p \tag{6}$$

$$= \underbrace{\sum_{(a,i,j)\in T} \log(1 + \exp(1 + d(\mathbf{Z}_{a,:}, \mathbf{Z}_{i,:}) - d(\mathbf{Z}_{a,:}, \mathbf{Z}_{j,:})))}_{f(\mathbf{Z})} + \sum_{i=1}^{\min\{N,d'\}} \lambda g[\sigma_i(\mathbf{Z})]. \tag{7}$$

There are four assumptions we need to satisfy to apply the general result from (Sun et al., 2017).

**A1.** $f$ is differentiable and has a Lipschitz gradient:

This stems from smoothing the triplet loss with the softplus function. The composition makes $f$ differentiable everywhere except at degenerate collapse points (which do not arise with practical initializations). The log-sum-exp structure ensures (locally) Lipschitz gradients (Chen et al., 2020).

**A2.** $g$ is concave, nondecreasing, and Lipschitz:

$g(x) = x^p$ for $x > 0$ has these properties.

**A3.** $\Psi$ is coercive:

For our objective, suppose $\|\mathbf{Z}\|_F \to \infty$. Then the sum of squared singular values diverges, so at least one $\sigma(\mathbf{Z}) \to \infty$. As $g(\mathbf{Z})$ is the sum of all $\sigma(\mathbf{Z})^p$, and $p > 0$, we therefore have $g(\mathbf{Z}) \to \infty$ and thus $\Psi(\mathbf{Z}) \to \infty$.

**A4.** $\Psi$ has the Kurdyka Lojasciewicz (KL) property:

As established in (Bolte et al., 2010), sums of o-minimal (definable) functions, such as our loss and regularizer, possess the KL property.

With all required assumptions satisfied, Theorem 1 of (Sun et al., 2017) applies and guarantees:

$$\sum_{k=1}^{\infty} \|\mathbf{Z}^{k+1} - \mathbf{Z}^k\|_F < \infty. \tag{8}$$

Therefore, the LORE algorithm converges to a stationary point. $\qquad\square$

## B    OPERATIONAL DETAILS FOR LORE

The optimization algorithm used for LORE is an adaptation of the original algorithm from (Sun et al., 2017).

The function takes the initialized embedding $\mathbf{Z}^0$, the regularization parameter $\lambda$, the Lipschitz constant of $\nabla f(\cdot)$, $\mu$, and the tolerance for convergence tol. The exact Lipschitz constant of $f(\cdot)$ is not known but was be empirically estimated to be strictly greater than $0.013$ by the Power Iteration method. Therefore, we set $\mu = 0.1$ throughout our experiments. The algorithm initializes the ordinal embedding and iteratively updates it by minimizing the smoothed ordinal loss plus Schatten-p regularization. Each step performs a proximal gradient update and singular value thresholding, repeating until convergence. Based on prior literature in the Schatten-p quasi-norm optimization literature (Wang et al., 2024; Sun et al., 2017; Lu et al., 2014), we fix $p = 0.5$ as it has been shown to have good empirical results across various applications. Both $\mu$ and $p$ are hyperparameters that are needed to be tuned for a practitioner. Moreover, though $\lambda$ is a clear hyperparameter, our empirical results show that a setting of $\lambda = 0.01$ works well across all of our experiments and we would suggest a practitioner start with this value. The tolerance tol is set to $10^{-5}$ throughout our experiments. Moreover, we cap the number of iterations to 1000 for all experiments to ensure reasonable runtimes. If longer runtimes are acceptable, this cap can be increased to improve performance. However, this is usually very marginal.

If we consider the most standard operational setting, i.e. $d' < N \ll T$, then the time complexity of each iteration is $\mathcal{O}(d'(T + Nd'))$. The dominating terms here are the number of percepts and the number of triplets used with the most intensive operation is in line 8 where the gradient of $f$ is calculated and a singular value decomposition is subsequently performed. As a result, LORE is scalable for higher $N$ and $d'$. One does need to be careful as $T$ could scale with $\mathcal{O}(N^3)$ if many triplets are chosen which could slow down each iteration.

## C  ADDITIONAL PLOTS FOR REGULARIZATION OF LORE

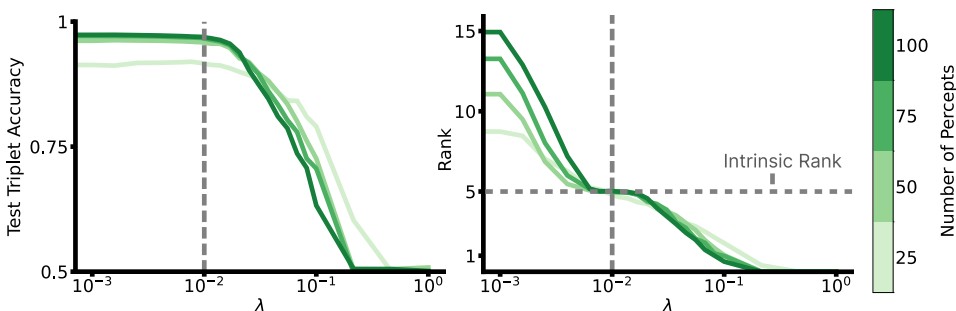

Figure 6: **LORE has high test triplet accuracy and intrinsic rank recovery across various number of percepts**. (Left) Mean test triplet accuracy vs $\lambda$ for LORE as number of percepts varies. (Right) Mean measured rank vs $\lambda$ for LORE as number of percepts varies.

Figure 6 shows the test triplet accuracy and intrinsic rank recovery of LORE as the number of percepts varies. We see that with greater number of percepts rank recovery stays roughly constant whereas the test triplet accuracy increases significantly from 25-50 percepts. Baseline parameters are intrinsic rank = 5, fraction of queries = 0.1, noise = 0.1.

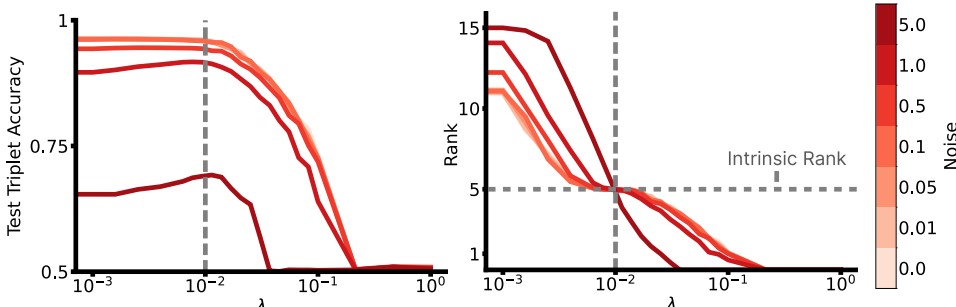

Figure 7: **LORE has high test triplet accuracy and intrinsic rank recovery across various noise levels**. (Left) Mean test triplet accuracy vs $\lambda$ for LORE as noise varies. (Right) Mean measured rank vs $\lambda$ for LORE as noise varies.

Figure 7 shows the test triplet accuracy and intrinsic rank recovery of LORE as the noise varies. We see that with greater noise rank recovery and test triplet accuracy both decrease. There is a dramatic drop in test triplet accuracy from 1 to 5 noise. Baseline parameters are intrinsic rank = 5, number of percepts = 50, fraction of queries = 0.1.

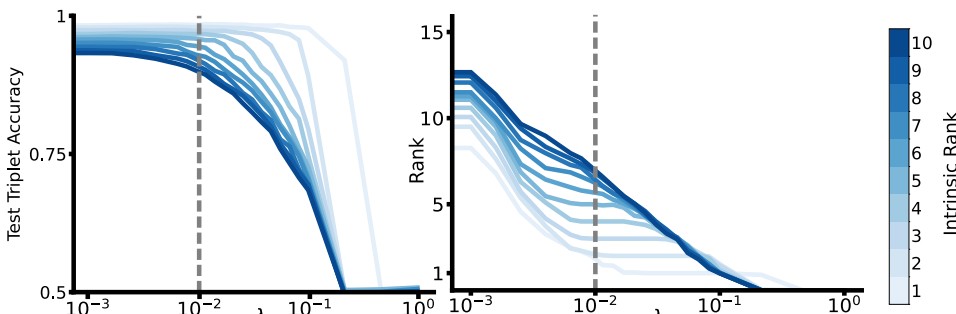

Figure 8: **LORE has high test triplet accuracy and intrinsic rank recovery across various intrinsic ranks**. (Left) Mean test triplet accuracy vs $\lambda$ for LORE as intrinsic rank varies. (Right) Mean measured rank vs $\lambda$ for LORE as intrinsic rank varies.

Figure 8 shows the test triplet accuracy and intrinsic rank recovery of LORE as the intrinsic rank varies. We see that for the same regularization level till approximately 8 dimensions, LORE can recover the rank. Baseline parameters are number of percepts = 50, fraction of queries = 0.1, noise = 0.1.

These results, together with Figure 2, show that LORE is quite robust to the various knobs (noise, number of percepts, intrinsic rank and number of queries) that can be tuned for OE applications and that a regularization setting of $\lambda = 0.01$ learns an embedding with both high triplet accuracy yet recover the intrinsic rank.

# D  ADDITIONAL PLOTS COMPARING LORE TO BASELINES

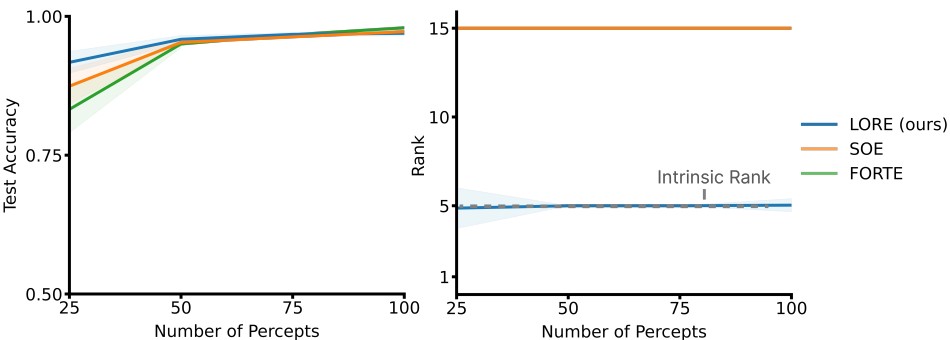

Figure 9: **Only LORE can recover the intrinsic rank while maintaining comparable test triplet accuracy**. (Left) Mean test triplet accuracy vs number of percepts used for LORE and the baselines. (Right) Mean measured rank vs number of percepts used for LORE and the baselines.

Figure 9 shows the test triplet accuracy and intrinsic rank recovery of LORE and the baselines as the number of percepts varies. We see that with greater number of percepts rank recovery stays roughly constant, though spread decreases, for LORE from 25-50 percepts. Baselines again cannot recover the intrinsic rank at all. Test triplet accuracy increases from 25-50 percepts for all OE algorithms. Baseline parameters are intrinsic rank = 5, fraction of queries = 0.1, noise = 0.1.

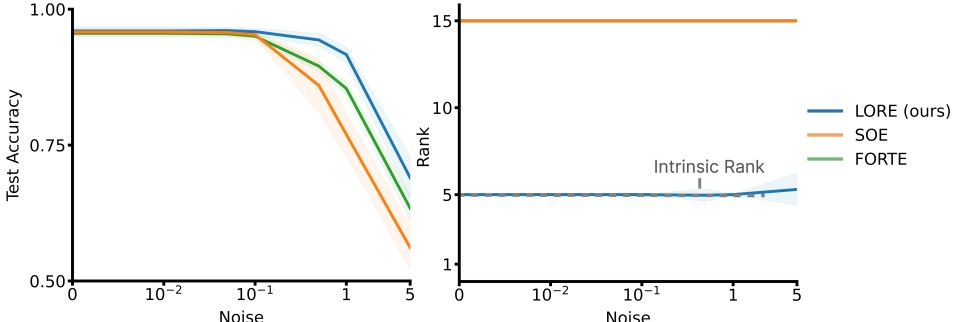

Figure 10: **Only LORE can recover the intrinsic rank while maintaining comparable test triplet accuracy**. (Left) Mean test triplet accuracy vs noise used for LORE and the baselines. (Right) Mean measured rank vs noise used for LORE and the baselines.

Figure 10 shows the test triplet accuracy and intrinsic rank recovery of LORE and the baselines as the noise varies. We see that with greater noise, LORE is still able to recover the intrinsic rank though spread increases with noise from 1-5. The baselines cannot recover the intrinsic rank at all. Test triplet accuracy decreases with noise for all OE algorithms though LORE still performs the best. Baseline parameters are intrinsic rank = 5, number of percepts = 50, fraction of queries = 0.1.

# E   CROWDSOURCED DATASET DETAILS

Table 3: Details of Crowdsourced Datasets Used

| Datasets | Num. Percepts | Num. Triplets | Triplet Type | Notes |
|---|---|---|---|---|
| Food-100 (Wilber et al., 2014) | 100 | 190,376 | Compared to A, which is more similar, B or C? | Images of foods. Converted to similarity triplets using 'cblearn' (Künstle & von Luxburg, 2024). |
| Materials (Lagunas et al., 2019) | 100 | 22,801 | Compared to A, which is more similar, B or C? | Images of materials. Converted to similarity triplets using 'cblearn' (Künstle & von Luxburg, 2024). |
| Cars (Kleindessner & Von Luxburg, 2017) | 68 | 7,097 | Which of A, B, C is the most central? | Images of cars. Central triplets converted to similarity triplets using 'cblearn' (Künstle & von Luxburg, 2024). |
| Musicians (Ellis et al., 2002) | 448 | 118,263 | Compared to A, which is more similar, B or C? | Names of musicians. Converted to similarity triplets using 'cblearn' (Künstle & von Luxburg, 2024). |

Of these datasets, the Cars dataset is known to be very noisy (Kleindessner & Von Luxburg, 2017; Vankadara et al., 2023). Food-100 has been used as a dataset to evaluate active querying methods (Canal et al., 2020). Musicians is known to be very undersampled in terms of triplets compared to the number of percepts and is not the desired operational setting of this work. A detailed characterization of the datasets is seen in Table 3.

## F    ADDITIONAL INTERPRETABILITY PLOTS

In this section we include interpretability plots for the other ordinal embedding methods on the Food-100 dataset. These plots are analogous to Figure 5 in the main paper. The procedure that we use to obtain these axes is to take the best performing ordinal embedding learned by each of these methods from Table 2 and then perform a principal component analysis (PCA) on the embedding to get the top three principal components. The reason for doing so is because ordinal embeddings cannot learn the directions of highest variance but only can learn an embedding that may be rotated, scaled or translated compared to the true perceptual space (Vankadara et al., 2023; Jain et al., 2016). Then, we vary the value of each principal component from its minimum to maximum value in the embedding and plot datapoints across each axis separately to see if there is any semantic meaning to the axis.

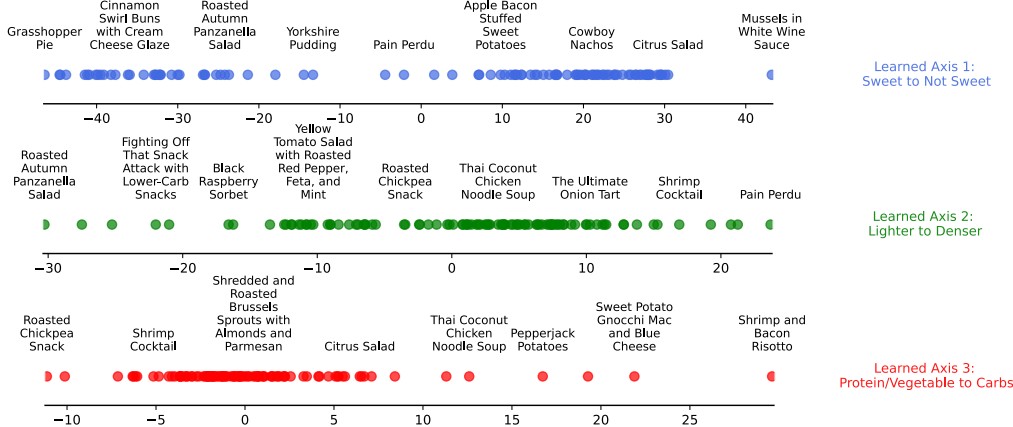

Figure 11: **CKL's learned axes are semantically interpretable**: Food groups as axis value varies for the first three learned axes of the CKL embedding learned on the Food-100 dataset. These are very similar to LORE's as seen in Figure 5 (Same embedding as one learned for Table 2).

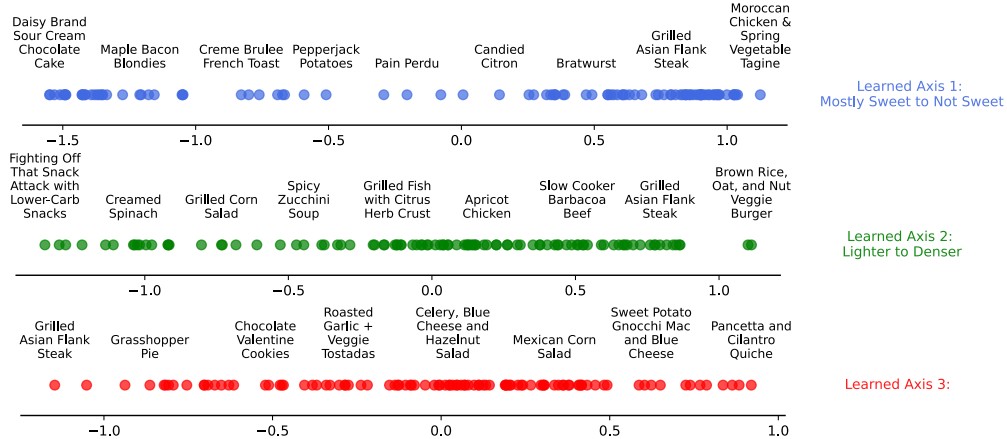

Figure 13: **FORTE's learned axes are not fully semantically interpretable**: Food groups as axis value varies for the first three learned axes of the FORTE embedding learned on the Food-100 dataset. Only the first two dimensions are semantically interpretable but the third is not compared to LORE's Figure 5 (Same embedding as one learned for Table 2).

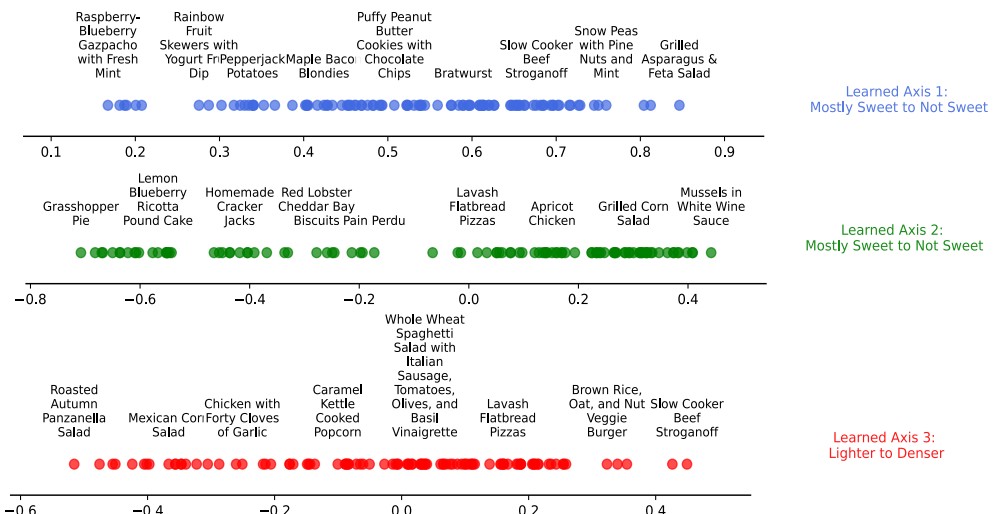

Figure 12: **SOE's learned axes are semantically interpretable but not parsimonious**: Food groups as axis value varies for the first three learned axes of the SOE embedding learned on the Food-100 dataset. These contain two dimensions that LORE does but not all Figure 5 (Same embedding as one learned for Table 2).

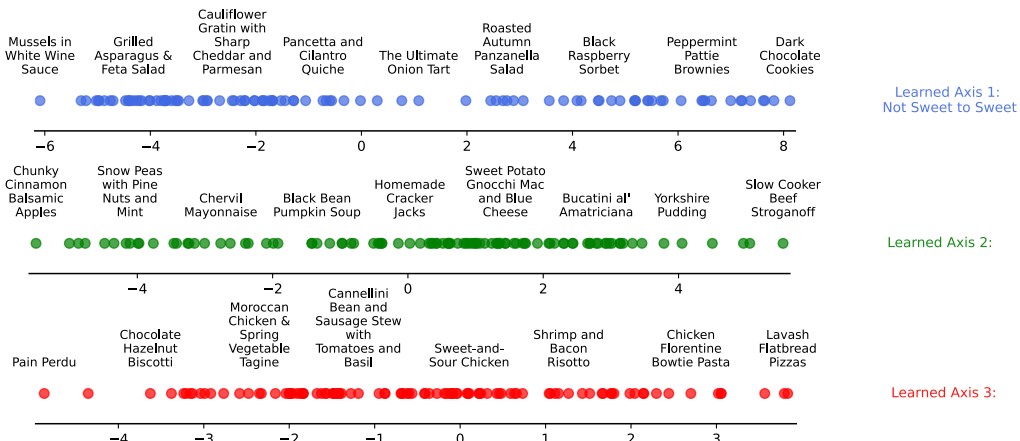

Figure 14: **t-STE's learned axes are not fully semantically interpretable**: Food groups as axis value varies for the first three learned axes of the t-STE embedding learned on the Food-100 dataset. Only the first dimension is semantically interpretable but the second and third are not compared to LORE's Figure 5 (Same embedding as one learned for Table 2).

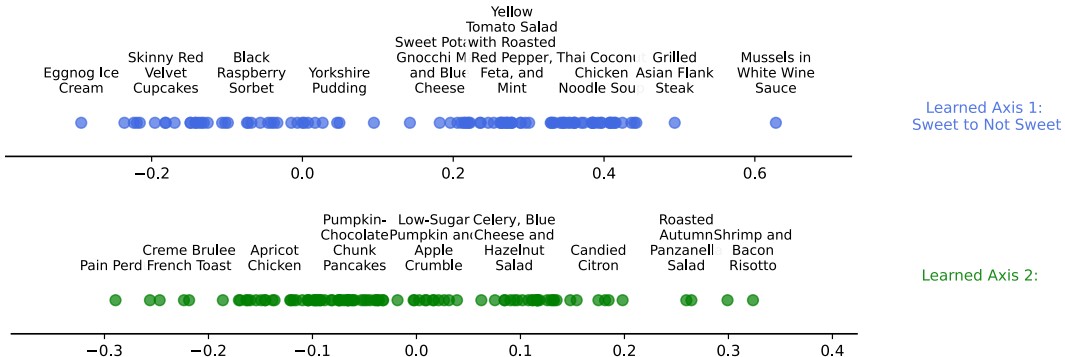

Figure 15: **Dim-CV's learned axes are not fully semantically interpretable**: Food groups as axis value varies for the first two learned axes of the Dim-CV embedding (chosen after retraining and dimensionality selection) learned on the Food-100 dataset. Only the first dimension is semantically interpretable but the second is not compared to Figure 5 (Same embedding as one learned for Table 2). Note that Dim-CV only learns two dimensions so we only show two axes here.

From the above plots, we see that only CKL is able to learn all the three interpretable dimensions that LORE is able to learn. Both FORTE and TSTE are able to learn one or two interpretable dimensions but not all three. While SOE does learn all three interpretable dimensions, it splits sweetness to not sweet across two dimensions rather than one as LORE and CKL do. Dim-CV meanwhile though it learns a low rank embedding (only two dimensions), the second axis is not interpretable at all. Moreover, as seen in Table 2, it takes an order of magnitude longer to train. Though both CKL and LORE are able to learn comparable interpretable dimensions, LORE is superior as it also learns a lower rank representation as well indicating that it does capture only the most important dimensions of the perceptual space without underfitting or overparameterizing.

## G    EXAMINING THE EFFECT OF $p$ ON LORE

In the following results we examine the effect of the $p$ in the ordinal embedding and rank recovery for LORE. Specifically, we vary $p$ with three values: 0.1, 0.5 (what we use in the paper) and 1 (equivalent to the nuclear norm). Like in C, we vary the number of percepts, noise and intrinsic rank while keeping the other parameters constant to see how $p$ affects the test triplet accuracy and rank recovery albeit at a reduced scale due to the computational cost of experiments. We include code for this experiment despite the additional time needed to run it. We find that $p = 1$ has quite similar results to $p = 0.5$ albeit with a reduced optimal regularization level of around $\lambda = 10^{-2.5}$. However, we notice that the leeway to choose the regularization parameter is much higher in absolute terms for $p = 0.5$ than $p = 1$ as the performance of $p = 1$ drops off more sharply as we deviate from the optimal regularization level. This is especially apparent for high noise levels. $p = 0.1$ however is essentially useless to obtain lower rank solutions, at least for the range of regularization that we have examined. The baseline parameters we use for all of them, unless we are varying them, are number of percepts = 50, intrinsic rank = 5, fraction of queries = 0.1 and noise = 0.1.

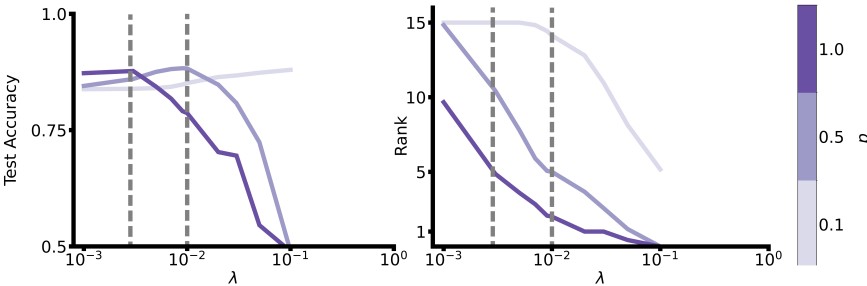

Figure 16: **Both $p = 0.5$ and $p = 1.0$ have stable rank recovery settings.**. (Left) Mean test triplet accuracy vs $\lambda$ for LORE. (Right) Mean measured rank vs $\lambda$ for LORE.

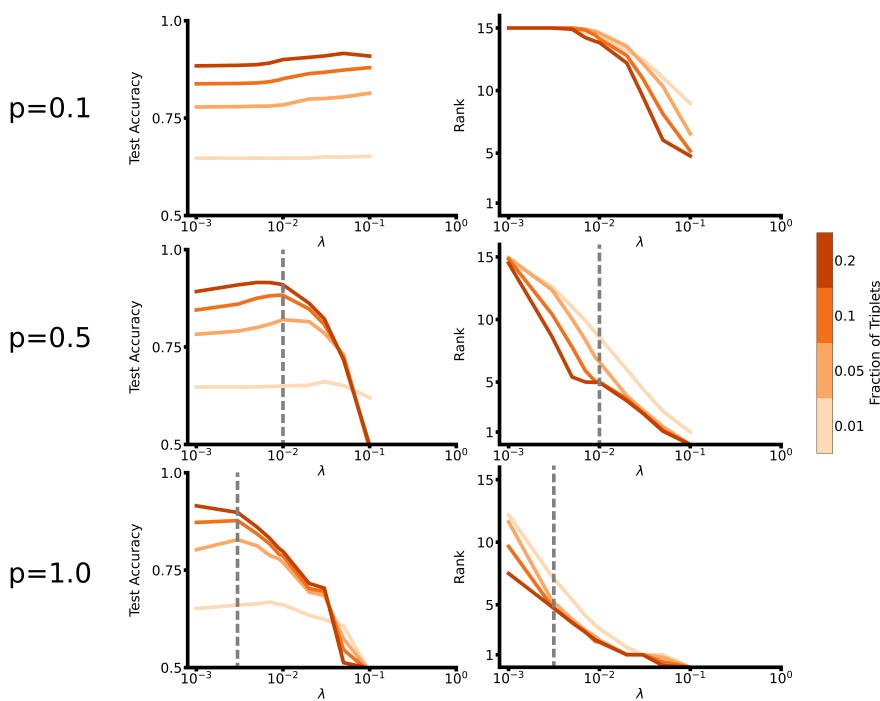

Figure 17: $p = 0.5$ **has the most stable rank recovery settings as fraction of train triplets varies**. (Left) Mean test triplet accuracy vs $\lambda$ for LORE as fraction of triplets used varies. (Right) Mean measured rank vs $\lambda$ for LORE as fraction of triplets used varies.

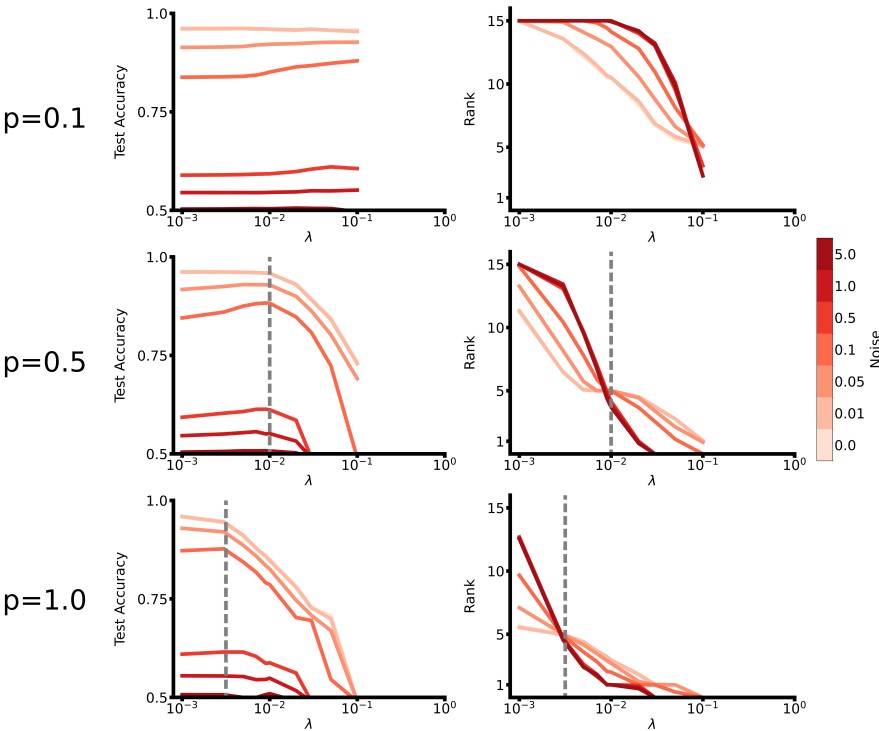

Figure 20: **Both $p = 0.5$ and $p = 1.0$ have stable rank recovery settings as noise varies**. (Left) Mean test triplet accuracy vs $\lambda$ for LORE as noise varies. (Right) Mean measured rank vs $\lambda$ for LORE as noise varies.

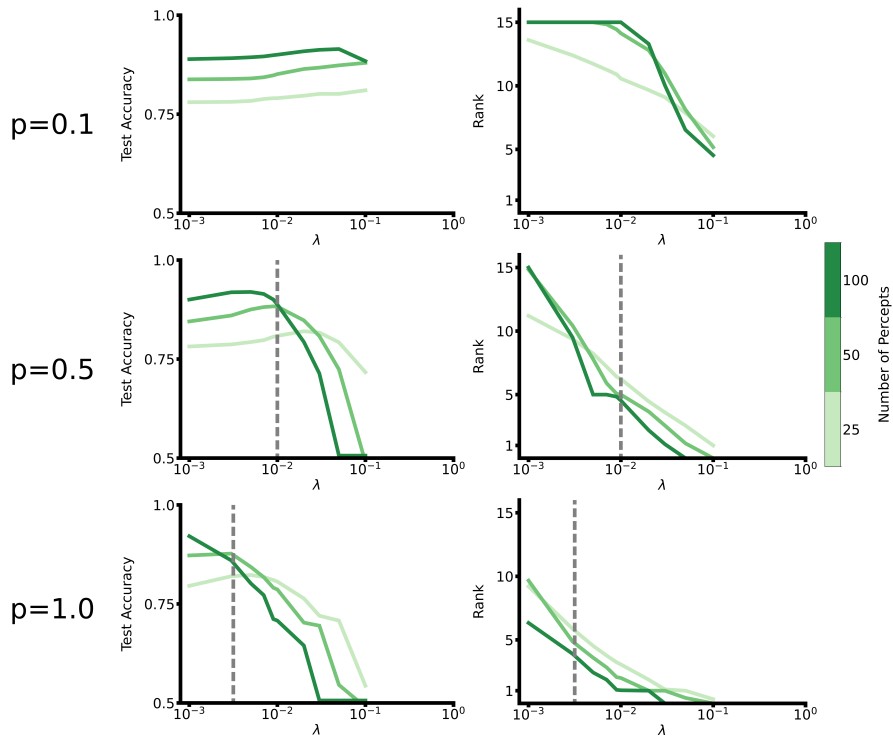

Figure 18: $p = 0.5$ **has the most stable rank recovery settings as number of percepts varies**. (Left) Mean test triplet accuracy vs $\lambda$ for LORE as number of percepts varies. (Right) Mean measured rank vs $\lambda$ for LORE as number of percepts varies.

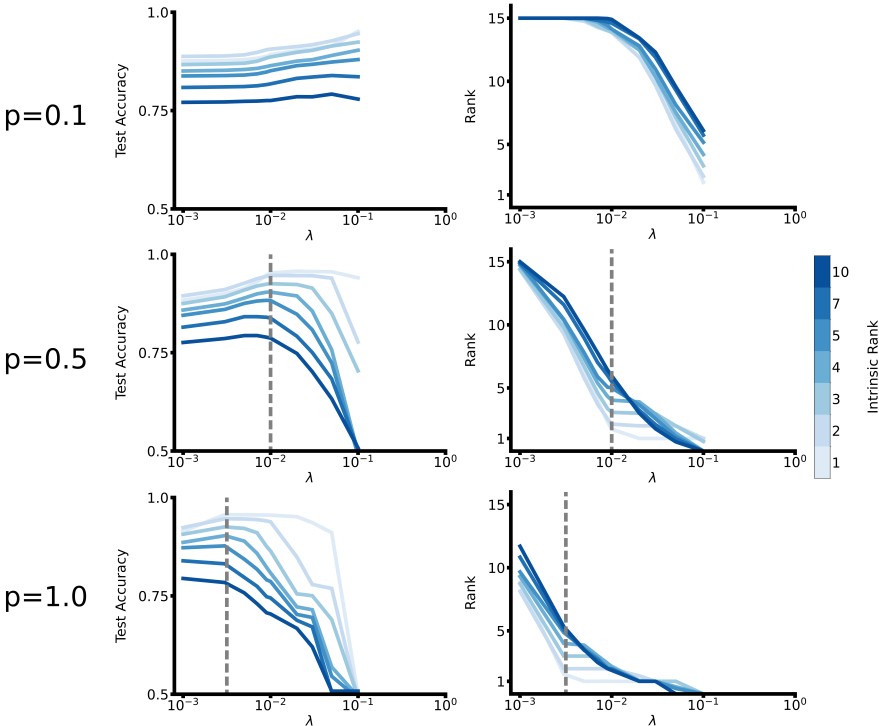

Figure 19: **Both** $p = 0.5$ **and** $p = 1.0$ **have stable rank recovery settings as intrinsic rank varies**. (Left) Mean test triplet accuracy vs $\lambda$ for LORE as intrinsic rank varies. (Right) Mean measured rank vs $\lambda$ for LORE as intrinsic rank varies.

## H EXPERIMENTAL SETUP FOR SECTION 5.2 AND SECTION 5.3

This experiment was performed on a SLURM server with over 30 GPUs of varying quality and compute power. We do not include the scripts used to run those experiments as they are highly complex due to parallelism and take too long to run (over 8 days). However, a quick rundown of the experiment is given below.

A grid search over all the following parameters was performed for these experiments. The grid search was performed in parallel over 30 GPUs. Each experiment was run for 30 runs with different random seeds. The results were averaged over the 30 runs and the standard deviation was calculated.

In our experiments, the various knobs we tune are as follows.

- **Number of Percepts** ($N$): We vary it from `[25, 50, 75, 100]` and use 50 Percepts as a default. We do not increase the number of percepts beyond 100 as the number of queries increases combinatorially. Additionally, this is not a practical number of percepts to collect for perceptual experiments.
- **True Dimension** ($d$): We vary it from `[1, 2, 3, 4, 5, 6, 7, 8, 9, 10]` and use 5 as a default. We do not examine over 10 dimensions as it is not possible to resolve that many dimensions without increasing the number of percepts due to the curse of dimensionality (Bishop & Nasrabadi, 2006).
- **Fraction of Queries used**: we vary it from `[0.1, 0.2, 0.3, 0.4, 0.5, 0.6, 0.7, 0.8, 0.9]` and use 0.1 as a default.
- **Noise** ($\sigma^2$): we vary it from `[0, 0.01, 0.05, 0.1, 0.5, 1.0, 5.0]` and use 0.1 as a default.
- **Regularization** ($\lambda$): This is only for LORE but we vary it with `[0, 0.001, 0.00158489, 0.00251189, 0.00398107, 0.00630957, 0.00768625, 0.00936329, 0.01, 0.01140625, 0.01389495, 0.01692667, 0.02061986, 0.02511886, 0.0305995, 0.03727594, 0.0454091, 0.05531681, 0.06738627, 0.08208914, 0.1, 0.21544347, 0.46415888, 1.]` and use 0.01 as a default.
- **Embedding Dimension** ($d'$): This is the dimension of the embedding we are trying to learn. This is only for the baselines other than LORE. We vary it from `[1, 2, 3, 4, 5, 6, 7, 8, 10, 12, 15]` and use 15 as a default.

The metrics we measure are as follows

- **Test Triplet Accuracy**: The accuracy of the test triplets on the test set. This is the main metric we use to measure performance.
- **Measured Rank**: The rank of the embedding matrix. This is a measure of how well the algorithm is able to recover the intrinsic rank of the data. We measure this by taking the SVD of the embedding matrix and counting the number of non-zero singular values. Specifically, we use the `rank` function from the `numpy` library to compute the rank of the embedding matrix.
- **Peak Signal to Noise Ratio**: The PSNR is a measure of the quality of the recovered matrix. However, note that the recovered embedding matrix has to be aligned to the true percepts matrix to compute the PSNR. The specific formulation is described in Appendix N.
- **Normalized Procrustes Distance**: The NPD is a measure of how well the recovered matrix matches the true matrix up to rotation, scaling and translation. To perform procrustes analysis, true percepts $\mathbf{P} \in \mathbb{R}^{N \times d}$ and the computed embedding $\mathbf{Z} \in \mathbb{R}^{N \times d'}$ must be the same shape. Therefore, we use the same subspace alignment technique to ensure that the two matrices have the same shape. The specific formulation is described in Appendix N.

It should be noted that test triplet accuracy and measured rank are the main metrics we use to measure performance as the other metrics require knowledge of the percepts $\mathbf{P}$ which is not known in practice.

## I EXPERIMENTAL SETUP FOR SECTION 5.4

50 random foods were chosen from the Food-100 dataset (Wilber et al., 2014). This was run on a server with 1 RTX3080 GPU and 128 GB of RAM. The names of the specific percepts are as follows.

```
['Cinnamon Swirl Buns with Cream Cheese Glaze',
 'Shrimp and Bacon Risotto',
 'Shrimp Cocktail',
 'Homemade Cracker Jacks',
 'Creme Brulee French Toast',
 'Red Lobster Cheddar Bay Biscuits',
 'Apple Bacon Stuffed Sweet Potatoes',
 'Sweet-and-Sour Chicken',
 'Pumpkin-Chocolate Chunk Pancakes',
 'Chocolate Hazelnut Biscotti',
 'Eggnog Ice Cream',
 'Celery, Blue Cheese and Hazelnut Salad',
 'Shredded and Roasted Brussels Sprouts with Almonds and Parmesan',
 'Low-Sugar Pumpkin and Apple Crumble',
 'Roasted Sweet Potatoes Recipe with Double Truffle Flavor and Parmesan',
 'Chicken Florentine Bowtie Pasta',
 'White Whole Wheat Pizza Dough',
 'Chervil Mayonnaise',
 'Pork Tenderloin in Tomatillo Sauce',
 'Yellow Tomato Salad with Roasted Red Pepper, Feta, and Mint',
 'Daisy Brand Sour Cream Chocolate Cake',
 'Shredded Brussels Sprouts & Apples',
 'Mexican Corn Salad',
 'Potato Skins',
 'Caramel Kettle Cooked Popcorn',
 'Roasted Garlic + Veggie Tostadas',
 'Pan Seared Scallops with Baby Greens and Citrus Mojo Vinaigrette',
 'Lemon Cranberry Scones',
 'Warm Butternut and Chickpea Salad with Tahini Dressing',
 'Fighting Off That Snack Attack with Lower-Carb Snacks',
 'Chicken with Forty Cloves of Garlic',
 'Edna Mae's Sour Cream Pancakes',
 'Sweet Potato Gnocchi Mac and Blue Cheese',
 'Yorkshire Pudding',
 'Luscious Lemon Squares',
 'Japanese Pizza',
 'Grilled Asparagus & Feta Salad',
 'Grilled Corn Salad',
 'Garlic Meatball Pasta',
 'Roasted Autumn Panzanella Salad',
 'Coconut Marinated Pork Tenderloin',
 'Black Raspberry Sorbet',
 'Mini Whole Wheat BBQ Chicken Calzones',
 'Mussels in White Wine Sauce',
 'Brown Rice, Oat, and Nut Veggie Burger',
 'Dark Chocolate Cookies',
 'Citrus Salad',
 'Roasted Carrots & Parsnip Puree',
 'South African Cheese, Grilled Onion & Tomato Panini (Braaibroodjie)',
 'Pinto Bean Salad with Avocado, Tomatoes, Red Onion, and Cilantro']
```

These names are passed to the SBERT library (Reimers & Gurevych, 2019) with the "all-mpnet-base-v2" model to get a 768 dimensional LLM embedding. To simulate various possible intrinsic

ranks, we use the truncated singular value decomposition to constrain the "true" perceptual representations of foods to intrinsic ranks 1-10. Specifically, the truncated SVD is the following.

The singular value decomposition of a matrix $\mathbf{P}' \in \mathbb{R}^{N \times d}$ is given by

$$\mathbf{P}' = \mathbf{U}\mathbf{\Sigma}\mathbf{V}^T \tag{9}$$

Here $\mathbf{U} \in \mathbb{R}^{N \times N}$, $\mathbf{S} \in \mathbb{R}^{N \times 768}$ and $\mathbf{V} \in \mathbb{R}^{768 \times 768}$. $\mathbf{U}$ and $\mathbf{V}$ have orthonormal columns and $\Sigma$ is a diagonal matrix with singular values $\sigma_1 \geq \sigma_2 \geq \cdots \geq \sigma_N > 0$. The intrinsic rank of the matrix is the number of non-zero singular values, which in this case is $N$ before truncation. We can truncate the SVD to fix an intrinsic rank of $d$.

If $\mathbf{U}_d \in \mathbb{R}^{N \times d}$ and $\mathbf{V}_d \in \mathbb{R}^{768 \times d}$ are the first $d$ columns of $\mathbf{U}$ and $\mathbf{V}$ respectively, $\Sigma_d \in \mathbb{R}^{d \times d}$ with the biggest $d$ singular values in the diagonal entries and otherwise 0 then we can write the truncated SVD of $\mathbf{P}'$ to get the our "true" perceptual representation of the foods as

$$\mathbf{P}' = \mathbf{U}_d\mathbf{\Sigma}_d\mathbf{V}_d^T \tag{10}$$

Then, we query just 5% of the total possible triplets (2940 out of a possible 58800) with 0.1 variance Gaussian noise added to the triplet comparisons to simulate the noise that humans have when answering triplet queries. 5% of the total queries is a reasonable setting common to most perceptual scaling experiments as it is usually the bare minimum of queries needed to fit a good embedding (Künstle et al., 2022; Vankadara et al., 2023). We repeat this sampling and query answer simulation process thirty times, independently with various random seeds, and train all of the the various OE algorithms. This is to obtain more robust results and avoid the effects of bad initializations or pathological training sets. The metrics we measure are as follows.

- **Test Triplet Accuracy**: The accuracy of the test triplets on the test set (fixed at 3000 queries not in the train set and chosen at random). This is the main metric we use to measure performance.

- **Measured Rank**: The rank of the embedding matrix. This is a measure of how well the algorithm is able to recover the intrinsic rank of the data. We measure this by taking the SVD of the embedding matrix and counting the number of non-zero singular values. Specifically, we use the `rank` function from the `numpy` library to compute the rank of the embedding matrix.

- **Peak Signal to Noise Ratio**: The PSNR is a measure of the quality of the recovered matrix. However, note that the recovered embedding matrix has to be aligned to the true percepts matrix to compute the PSNR. The specific formulation is described in Appendix N. (We do not report these in the paper)

- **Normalized Procrustes Distance**: The NPD is a measure of how well the recovered matrix matches the true matrix up to rotation, scaling and translation. To perform procrustes analysis, true percepts $\mathbf{P} \in \mathbb{R}^{N \times d}$ and the computed embedding $\mathbf{Z} \in \mathbb{R}^{N \times d'}$ must be the same shape. Therefore, we use the same subspace alignment technique to ensure that the two matrices have the same shape. The specific formulation is described in Appendix N. (We do not report these in the paper)

Code for this experiment is included in the supplemental material.

## J    Experimental Setup for Section 5.5

This was run on a server with 1 RTX3080 GPU and 128 GB of RAM.

For LORE, we set the regularization parameter, $\lambda$, to 0.01. For all OE methods, we set the number of dimensions of the OE, $d'$, to 15.

Note that for this experiment unlike in Appendix I, we do not have access to the true percepts $\mathbf{P}$ and therefore cannot compute the PSNR or NPD. We only report the test triplet accuracy and measured rank. For Dim-CV we use a total of 5 cross validation folds but do not use multiple initializations due to computational constraints. As it is, Dim-CV is already two orders of magnitude slower than all of the OE algorithms due to the additional burden of training multiple embeddings due to the cross validation procedure. From our observations, increasing the number of cross validation folds and using multiple initializations reduces the standard deviation of the Dim-CV test triplet accuracy and rank but not the mean. Time however, increases linearly with the number of cross validation folds and initializations.

Code for this experiment is included in the supplemental material.

# K    ADDITIONAL CROWDSOURCED DATASET EXPERIMENTS

The results on one more crowdsourced real life dataset, the musicians dataset (Ellis et al., 2002) are seen in Table 4. This dataset contains 448 percepts (musicians) and 118,263 triplet comparisons collected from human annotators. This is considerably undersampled in terms of triplets compared to the other datasets included in the main text. For example, Food-100 contains more triplets even though it has fewer percepts. The results are shown in the figure below. We do not run the Dim-CV due to computational constraints from the hypothesis testing procedure. We anticipate it would take ~7000 seconds per iteration to run Dim-CV on this dataset given that it took ~1700 seconds per iteration on the Food-100 dataset which has roughly one fourth the number of percepts. Therefore, we exclude it from this experiment. We set $d' = 30$ as the embedding dimension for all OE methods as this has considerably more number of percepts than the other datasets.

Table 4: Comparison of OEs on Musicians Dataset

| Method | Musicians | | |
|---|---|---|---|
| Metric $\pm$ Std | Test Acc. | Rank | Time (s) |
| **LORE** (Ours) | $75.63 \pm 0.94$ | $\mathbf{27.8 \pm 0.55}$ | $13.82 \pm 9.72$ |
| **SOE** | $81.41 \pm 0.93$ | $30 \pm 0.0$ | $28.45 \pm 2.20$ |
| **FORTE** | $69.94 \pm 1.61$ | $30 \pm 0.0$ | $8.63 \pm 2.79$ |
| **t-STE** | $79.49 \pm 1.52$ | $30 \pm 0.0$ | $98.97 \pm 81.26$ |
| **CKL** | $78.05 \pm 0.96$ | $30 \pm 0.0$ | $24.3 \pm 10.51$ |

## L    SCALABILITY OF LORE

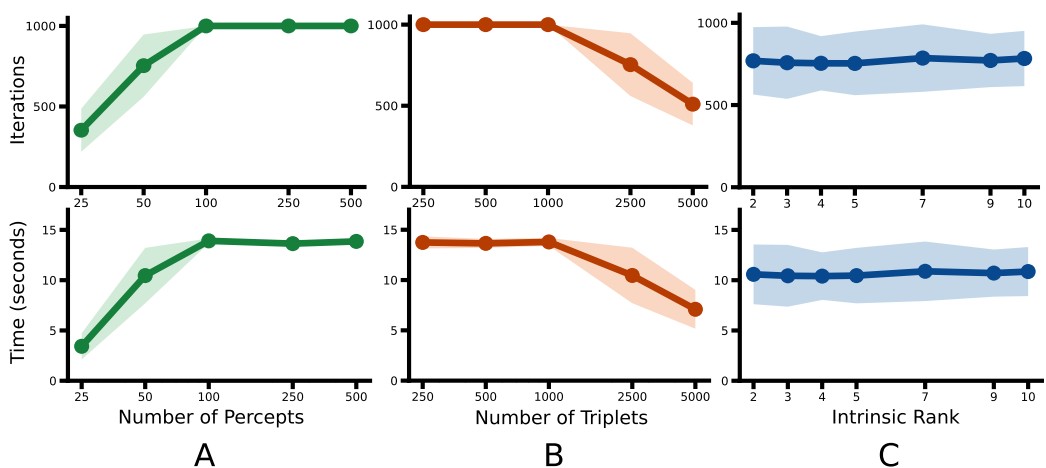

Figure 21: **LORE remains scalable as dataset parameters change**: (A) Time and Number of Iterations as Number of Percepts Varies (log scaled x axis) (B) Time and Number of Iterations as Number of Triplets Varies (log scaled x axis) (C) Time and Number of Iterations as Intrinsic Rank Varies. Error bars indicate $\pm$ two standard deviations over 30 random seeds of the generative process for different datasets. Baseline parameters are intrinsic rank = 5, number of percepts = 50, number of triplets = 2500.

For our empirical implementation in this paper, we cap the number of iterations of LORE to 1000. In this experiment, we fix the noise to a moderate noise of 0.1 variance sampled from a Gaussian distribution. As seen in Figure 21, which varies one dataset characteristic while keeping the others constant, the number of iterations correlates almost perfectly with the time taken to run LORE. Therefore, if LORE converges in fewer iterations, it will take less time to run and the length of each iteration is roughly constant for a given dataset. We see that as number of percepts increases, the number of iterations increases until it hits the 1000 iteration cap. This is likely due to the fact that as the number of percepts increases, the optimization problem becomes more ambiguous as $\mathbf{Z}$ increases in size. As the number of triplets increases, the number of iterations decreases likely because there are more constraints to guide the optimization. Finally, the intrinsic rank does not seem to have a significant effect on the number of iterations needed to converge.

## M  CONVERGENCE OF LORE

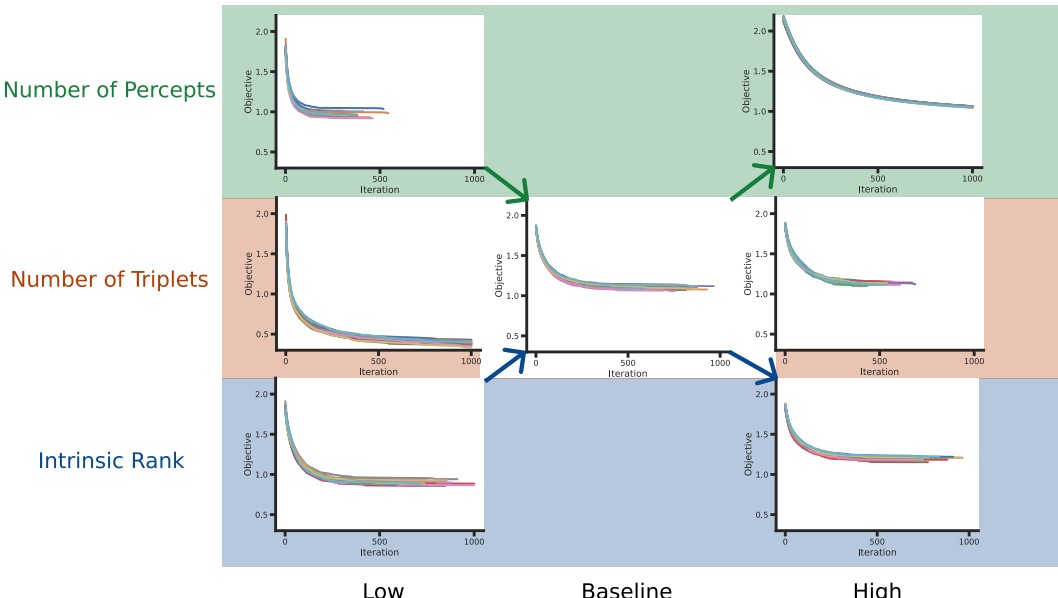

Figure 22: **LORE convergences smoothly despite different initializations and nonconvexity**: Objective function values as LORE trains for 30 random seeds for the same true perceptual space. Each color is a separate seed run. Number of Percepts, Number of Triplets and Intrinsic Rank are varied individually. Baseline parameters are a space defined by intrinsic rank = 5, number of percepts = 50, number of triplets = 2500.

To visually examine the convergence of LORE we run an experiment where we fix the true perceptual space and vary the random seed of the initialization of LORE. We see in Figure 22 that despite the non-convexity of the optimization problem, LORE converges smoothly across different random initializations. We vary one dataset characteristic at a time while keeping the others constant like in the previous section. We see that as the number of percepts increases (25, 50, 500), the objective function convergence takes longer likely due to the increased ambiguity of the optimization problem as the size of $\mathbf{Z}$ increases. As the number of triplets increases (250, 2500, 5000), convergence is faster likely due to the increased constraints on the optimization problem. Finally, the intrinsic rank (2, 5, 10) does not seem to have a significant effect on convergence speed.

Taken together with Appendix L, these results indicate that LORE smoothly converges despite the non-convexity of the optimization problem and that the number of iterations taken to converge scales reasonably with dataset parameters. This empirically shows the provable convergence to a stationary point that we show in Appendix A. This result in combination with the fact that local minima for ordinal embedding problems are often good solutions (Vankadara et al., 2023) indicates that LORE is a scalable and practical algorithm for ordinal embedding in real world settings and that the theory supports the empirical findings.

# N    FORMULATION OF OTHER METRICS

Code for all of these implementations is included in the supplemental material.

## N.1    SUBSPACE ALIGNMENT

To perform procrustes analysis, true percepts $\mathbf{P} \in \mathbb{R}^{N \times d}$ and the computed embedding $\mathbf{Z} \in \mathbb{R}^{N \times d'}$ must be the same shape.

Specifically we compute

$$\mathbf{P_c} = \mathbf{P} - \mathbf{1}_N \mu_{\mathbf{P}}^T \quad \text{and} \quad \mathbf{Z_c} = \mathbf{Z} - \mathbf{1}_N \mu_{\mathbf{Z}}^T \tag{11}$$

Then, we compute the tikhonov regularized projection matrix to prevent numerical instability due to ill conditioning. We use a regularization parameter of $\eta = 1e - 3$.

$$\mathbf{A} = (\mathbf{Z}_c^T \mathbf{Z}_c + \eta \mathbf{I}_d)^{-1} \mathbf{Z}_c^T \mathbf{P}_c \tag{12}$$

Then, we can get the aligned ordinal embedding $\mathbf{Z}_{\text{aligned}} = \mathbf{Z}_c \mathbf{A} + \mathbf{1}_N \mu_{\mathbf{P}}^T$.

## N.2    NORMALIZED PROCRUSTES DISTANCE

Now that we have an aligned matrix the same shape as $\mathbf{P}$, the normalized procrustes distance between the aligned embedding and the true percepts can be computed as

$$\text{Normalized Procrustes Distance} = \frac{\|\mathbf{P} - \mathbf{Z}_{\text{aligned}}\|_F}{\|\mathbf{Z}_c\|_F} \tag{13}$$

## N.3    PEAK SIGNAL TO NOISE RATIO

The Peak Signal to Noise Ratio (PSNR) is a measure of the quality of the recovered matrix and is defined as $20 \log_{10}(\max(\mathbf{Z}_{\text{aligned}})/(\|\mathbf{Z}_{\text{aligned}} - \mathbf{P}\|_F))$ where $\mathbf{P}$ is the true matrix.

## O   LLM USAGE

In this work, we leverage the use of large language models for two purposes. (1) to refine the writing by eliminating grammatical errors and improving flow. However, these were only used at the individual paragraph level rather than whole sections and (2) to discover similar papers during the literature review for the related work. Specifically, we searched for terms like "distance metric learning", "contrastive learning", "psychophysical scaling" etc.

