# OpenReview forum: "LORE: Jointly Learning The Intrinsic Dimensionality and Relative Similarity Structure from Ordinal Data"
_ICLR.cc/2026/Conference — ICLR 2026 Poster_

### Official Review · Reviewer_kWHX · 2025-11-01

**Soundness:** 3
**Presentation:** 2
**Contribution:** 2
**Rating:** 4
**Confidence:** 5

**Summary:**

This paper presents LORE, a framework that jointly learns ordinal embeddings and their intrinsic dimensionality using a nonconvex Schatten-p regularization and an iteratively reweighted optimization algorithm. Experiments on synthetic and real perceptual datasets show that LORE can automatically recover low-rank, interpretable embeddings with competitive triplet accuracy.

**Strengths:**

1. The paper presents an interesting framework (LORE) that aims to jointly learn ordinal embeddings and their intrinsic dimensionality, addressing a recognized limitation of existing approaches.

2. The proposed optimization procedure is clearly described and includes a convergence argument, suggesting technical soundness.

3. Experimental results across synthetic and real perceptual datasets provide encouraging evidence that LORE can recover compact and interpretable embeddings.

**Weaknesses:**

1. The paper lacks a theoretical analysis explaining under what conditions the Schatten-p regularization can correctly recover the intrinsic rank, which limits the strength of its main claim.

2. The iteratively reweighted optimization is presented formally but lacks practical insight; for example, the paper does not show convergence curves, runtime comparisons, or how initialization influences final embeddings.

3. The figures lack sufficient information for interpretation. Several plots (e.g., Figure 2 and Figure 3) omit axis labels or error ranges, and some results do not specify experimental settings or data sources, reducing the clarity and comparability of the findings.

4. The study provides limited discussion of hyperparameter sensitivity, particularly the effects of λ and p on performance and rank estimation.

**Questions:**

1. Under what data or noise conditions can the Schatten-p regularization reliably recover the intrinsic rank?

2. How sensitive is LORE to the choices of \lambda and p?

3. Could the authors show more evidence of the optimization’s empirical behavior, such as convergence stability or runtime?

4. All experiments are based on psychophysical or human-judgment data (food, music, cars). Have the authors tested, or plan to test, the method on non-perceptual ordinal datasets (e.g., image or text similarity)?

---

> ### Author Response · Authors · 2025-11-25
> **Official Comment by Authors to Reviewer kWHX**
>
> Thank you very much for the valuable feedback. Below we address your concerns. Please see the revised manuscript (changes in blue, to be removed for camera-ready).
>
> 1. **"Under what data or noise conditions can the Schatten-p regularization reliably recover the intrinsic rank?"**
>      - We agree that for low rank objectives data and noise characteristics can affect intrinsic rank recovery. However, for LORE we show that for most dataset characteristics and up to moderate noise, this is not a concern.
> 	     - In Section 5.2 and Appendix C, we empirically show how the various characteristics of the dataset including the number of queries used (Figure 2), number of percepts (Figure 6), noise (Figure 7) and intrinsic rank (Figure 8 newly added for the rebuttal) affect both the test triplet accuracy and the recovered intrinsic rank for LORE as $\lambda$ varies.
> 	     - For low to moderate noise (0-1 variance), LORE is stable with respect to $\lambda$, and recovers both the intrinsic rank and has high test triplet accuracy.
> 	     - In all of these settings a fixed regularization level of $\lambda=0.01$ allows us to learn the true perceptual space without underfitting or overparameterizing.
> 2. **"How sensitive is LORE to the choices of \lambda and p?"** and **"The study provides limited discussion of hyperparameter sensitivity, particularly the effects of λ and p on performance and rank estimation."**
> 	 - We agree that optimization methods are usually sensitive to hyperparameter choice. However, our experimental results show that this is not the case for LORE and that there exist a reasonably wide range of regularization values where LORE performs well.
> 		 - Here, $p$ is not a hyperparameter that a practitioner must tune, and is simply set a priori to $p=0.5$ as it has been shown to work well empirically in other Schatten Quasi Norm optimization settings, including image denoising and matrix completion (Sun et al., 2017; Wang et al., 2024).
> 		 - As seen in Section 5.2 and Appendix C, $\lambda =0.01$ consistently achieves both high triplet accuracy and good intrinsic rank recovery for various datasets.
> 		 - Therefore for typical perceptual experiments (intrinsic rank 1-10), enough triplets, low to moderate noise, and reasonable number of percepts, LORE is not very sensitive to the choice of $\lambda$
> 3. **"Could the authors show more evidence of the optimization’s empirical behavior, such as convergence stability or runtime?"** and **"The iteratively reweighted optimization is presented formally but lacks practical insight; for example, the paper does not show convergence curves, runtime comparisons, or how initialization influences final embeddings."**
> 	 - Thank you! We agree that evidence of the optimization's convergence strengthens our claims.
> 		 - We add Appendix K which examines how runtime and number of iterations vary with number of percepts, triplets and intrinsic rank and find that LORE scales reasonably with all of these.
> 			- Convergence time strongly correlates with number of iterations
> 			- As number of percepts increases, so do iterations (until 1000 iterations as we cap the number of iterations) as optimization likely becomes more complex.
> 			- As number of triplets increases, iterations decreases as there are more constraints on optimization.
> 			- Convergence time is agnostic to Intrinsic Rank
> 		 - We also add Appendix L which presents convergence curves for different initializations of LORE for the same perceptual space. In all cases, different initializations converge to very similar objective values, showing reliable optimization.
> 		 - Empirical results in Sections 5.3, 5.4 and 5.5 all include error bars/standard deviation from independent initializations and different train/test splits. These are small for both test triplet accuracy and recovered rank indicating stability and convergence of the LORE optimization method.
> 4. **"All experiments are based on psychophysical or human-judgment data (food, music, cars). Have the authors tested, or plan to test, the method on non-perceptual ordinal datasets (e.g., image or text similarity)?"**
> 	- Thank you! We appreciate the suggestion to consider non-perceptual ordinal tasks.
> 		- For non-human judgement datasets like text or image similarity, we believe better suited methods exist, like distance metric learning or contrastive learning which can leverage the presence of explicit representations like RGB images or tokenized text.
> 		- OEs however instead rely only on human judgement responses and are well suited when no explicit representation exists (or would like to avoid biasing learned embeddings based on existing representations)
> 		- When discriminability between percepts is the main goal, explicitly modeling such information is preferable. Thus, we do not include non-human judgement datasets here though similar Schatten Quasi-Norm optimization mechanisms may be interesting to examine.

---

> ### Author Response · Authors · 2025-11-25
> **Official Comments by Authors to Reviewer kWHX continued**
>
> 5. **"The figures lack sufficient information for interpretation. Several plots (e.g., Figure 2 and Figure 3) omit axis labels or error ranges, and some results do not specify experimental settings or data sources, reducing the clarity and comparability of the findings."**
> 	- Thank you! We appreciate your eye for detail and apologize for the oversight. We have fixed the relevant plots and add more details into all of the experimental sections with more detailed setup and appendix explanations.
>
>
> **References**
> - Wang et al., "Efficient Active Manifold Identification via Accelerated Iteratively Reweighted Nuclear Norm Minimization," JMLR, 2024.
> - Sun et al., "Convergence of proximal iteratively reweighted nuclear norm algorithm for image processing," IEEE TIP, 2017.
>
> Thank you very much for your detailed feedback which we have worked on to significantly strengthen our work.
>
> If we have answered your questions and have addressed your concerns, we kindly invite you to increase your score. If you still have any more questions, please do let us know and we will do our best to answer them.

---

### Official Review · Reviewer_inQi · 2025-11-01

**Soundness:** 3
**Presentation:** 3
**Contribution:** 3
**Rating:** 6
**Confidence:** 4

**Summary:**

The paper introduces LORE (LOw Rank Embedding), a novel and scalable framework designed to jointly learn the intrinsic dimensionality ($d$) and the optimal relative structure of perceptual spaces from noisy ordinal data (triplet comparisons of the form "A is more similar to B than C"). Addressing a fundamental limitation of existing Ordinal Embedding (OE) methods that rely on pre-defined or estimated dimensions, LORE leverages a low-rank constraint on the embedding matrix $Z$ and employs a highly non-convex Schatten quasi-norm as a regularizer to promote the discovery of the true intrinsic dimensionality. The optimization is handled by an effective iteratively reweighted algorithm with provable convergence guarantees. Extensive experiments on synthetic data, simulated perceptual spaces, and real-world crowdsourced datasets demonstrate that LORE successfully recovers the true intrinsic rank, achieves competitive triplet accuracy, and yields semantically interpretable embedding axes.

**Strengths:**

1. The paper addresses the critical and underexplored problem of jointly discovering the intrinsic dimensionality and relative structure in perceptual spaces, which is a key limitation of prior Ordinal Embedding (OE) methods. The introduction of the low-rank constraint via the non-convex Schatten quasi-norm is highly novel within the OE literature.

2. Unlike many empirical approaches, LORE provides a convergence theorem (Theorem 1, page 5) for its optimization objective and the proposed iterative reweighted algorithm. This rigorous theoretical foundation significantly strengthens the paper's contribution.

3. The experiments are thorough and persuasive. LORE successfully recovers the true intrinsic rank in synthetic and simulated LLM perceptual spaces (Figure 4) where other baselines fail. On real crowdsourced data (Food-100, Musicians, Cars), it maintains high triplet accuracy while achieving significantly lower rank embeddings compared to SOTA OE methods (Table 5).

4. The learned embedding axes (Figure 5) are shown to be semantically interpretable (e.g., "Sweet to Savory," "Learned Axis 1"), offering valuable insights into the underlying perceptual characteristics of the data, which is highly beneficial for discovery tasks.

**Weaknesses:**

1. While the paper provides a convergence theorem, the optimization objective $\min \Psi(Z)$ remains highly non-convex. The analysis primarily focuses on convergence to a stationary point, which may not always be the globally optimal solution. A more in-depth discussion on the practical robustness to initialization and the likelihood of escaping poor local minima would be beneficial.

2. The LORE objective function includes several regularization parameters ($\lambda, \tau, \mu$). Although Figure 2 demonstrates stability across a range of $\lambda$ values for a fixed $\tau$, a full exploration of the joint sensitivity of $\lambda$ and $\tau$ is absent. These parameters are crucial for balancing triplet accuracy and rank recovery, and their interplay needs more detailed investigation.

3. The paper should explicitly discuss the cases where the intrinsic rank $d$ may not be an integer (e.g., fractional dimensionality in complex manifold structures) and whether LORE's reliance on a rank constraint limits its ability to fully capture these more intricate data structures.

**Questions:**

1. The optimization relies on the non-convex Schatten quasi-norm. Could the authors provide a more detailed analysis or empirical evidence (e.g., through multiple restarts with different random initializations) showing the consistency and quality of the stationary points reached by the algorithm?

2. The paper uses a simulated perceptual space derived from a large language model (LLM) embedding. Can the authors provide more intuition or validation for why the LLM's embedding space represents a "true perceptual $d$-dimensional space" that LORE is attempting to recover, and how the inherent noise was modeled in this specific experiment?

3. Regarding the runtime complexity, how does the convergence speed (number of iterations for Algorithm 1) of LORE change as the total number of triplets ($T$) and the true intrinsic dimension ($d$) scale?

---

> ### Author Response · Authors · 2025-11-25
> **Official Comment by Authors to Reviewer inQi**
>
> Thank you very much for the valuable feedback. Below we address your concerns. Please see the revised manuscript (changes in blue, to be removed for camera-ready).
>
> 1. **"While the paper provides a convergence theorem, the optimization objective $\text{min}\Psi(Z)$  remains highly non-convex. The analysis primarily focuses on convergence to a stationary point, which may not always be the globally optimal solution. A more in-depth discussion on the practical robustness to initialization and the likelihood of escaping poor local minima would be beneficial."** and "**The optimization relies on the non-convex Schatten quasi-norm. Could the authors provide a more detailed analysis or empirical evidence (e.g., through multiple restarts with different random initializations) showing the consistency and quality of the stationary points reached by the algorithm?**"
> 	- We agree that non convex optimization raises questions about the quality of local minima. However, our experiments show that LORE, similar to other OE algorithms (Vankadara et al. 2023), does not suffer from poor local minima or bad initializations/convergence.
> 		- We show in Sections 5.3, 5.4, and 5.5, error bars/standard deviation (over 30 train-test splits) are consistently narrow, indicating stability; e.g., for the Food-100 dataset, LORE's std. dev. is only 0.27 at >82% accuracy.
> 		- Additionally, in a new experiment added to Appendix L, we plot convergence curves for different initializations while varying number of percepts, triplets, and intrinsic rank. Final objective values across initializations remain close, even if iteration counts vary by seed indicating that convergence is robust.
> 		- Both prior empirical work (Vankadara et al. 2023) and theory (Bower et al. 2018) suggest that OEs do not have bad local minima. This is a result we see in our empirical evaluation too.
> 		- These results, in combination with Theorem 1 (which proves that LORE converges to a stationary point), suggest that LORE consistently and reliably reaches high quality local minima.
> 2. **"The paper uses a simulated perceptual space derived from a large language model (LLM) embedding. Can the authors provide more intuition or validation for why the LLM's embedding space represents a "true perceptual -dimensional space" that LORE is attempting to recover, and how the inherent noise was modeled in this specific experiment?"**
> 	- Real datasets lack ground-truth intrinsic rank, so we create a simulated experiment with a known intrinsic rank. Recent work shows that Large Language Models (LLMs) have similar psychophysical similarities as humans (Marjieh et al., 2024). Therefore we use the LLM as a proxy for a human.
> 		- To create the "true" perceptual space with fixed intrinsic rank, we perform a Truncated Singular Value Decomposition (Truncated SVD) (dim 1-10) on the 784 LLM embedding to obtain a low rank approximation of the LLM embedding space.
> 		- The data generative process samples triplets, injects Gaussian noise to the distances between percepts (Is A closer to B or C? Process adds noise to dist(A,B) and dist(B, C)), which is identical to modelling human response noise in (Canal et al., 2022) (Terada et al., 2024) and train OEs for 30 independent initializations and train test splits.
> 		- Added additional explanations are added to Section 5.4 and Appendix H for better reproducibility.
> 3. **"The optimization relies on the non-convex Schatten quasi-norm. Could the authors provide a more detailed analysis or empirical evidence (e.g., through multiple restarts with different random initializations) showing the consistency and quality of the stationary points reached by the algorithm?"**
> 	- Thank you! We agree and add these additional results in Appendix K.
> 		- Convergence time strongly correlates with number of iterations
> 		- As number of percepts increases, so do iterations (until 1000 iterations as we cap the number of iterations). Optimization likely becomes more complex.
> 		- As number of triplets increases, iterations decreases. More constraints on optimization.
> 		- Convergence time agnostic to Intrinsic Rank
> 	- Also added Appendix L which shows that LORE is agnostic to initialization and reliably obtains good local minima

---

> ### Author Response · Authors · 2025-11-25
> **Official Comment by Authors to Reviewer inQi continued**
>
> 4. **"The LORE objective function includes several regularization parameters ($\lambda, \tau, \mu$). Although Figure 2 demonstrates stability across a range of $\lambda$ values for a fixed $\tau$ , a full exploration of the joint sensitivity of $\lambda$  and $\tau$ is absent. These parameters are crucial for balancing triplet accuracy and rank recovery, and their interplay needs more detailed investigation."**
> 	- We agree that for most optimization algorithms, hyperparameter tuning is critical. However, as we show in Sections 5.1 and 5.2 LORE only has one hyperparameter; the strength of the regularization $\lambda$.
> 		- For a practitioner, set $\lambda = 0.01$.
> 		- $\mu$ is not a hyperparameter and only needs to be greater than the Lipschitz constant of the triplet loss term. Empirically we found that $\mu >0.013$ so we set $\mu=0.1$ throughout our empirical experiments and we suggest the practitioner do the same. Details in Appendix B. It only affects convergence rate.
> 		- Figure 2 shows how for varying $T$, number of triplets, and $\lambda$ how the LORE performs with respect to test triplet accuracy and rank and that $\lambda=0.01$ consistently works well across $T$.
> 		- Further results varying other (dataset) parameters seen in Appendix C.
> 5. **"The paper should explicitly discuss the cases where the intrinsic rank  may not be an integer (e.g., fractional dimensionality in complex manifold structures) and whether LORE's reliance on a rank constraint limits its ability to fully capture these more intricate data structures."**
> 	- This is a very interesting question! In practice perceptual experiments are limited in number of percepts which makes fractional and higher dimensions harder to resolve.
> 	- As a result we assume only integral rank, which is standard for manifold theory. We clarify this in the background section.
>
> **References**
> - Vankadara et al., "Insights into ordinal embedding algorithms: A systematic evaluation," Journal of Machine Learning Research, 2023.
> - Marjieh et al., "Large language models predict human sensory judgments across six modalities," Scientific Reports, 2024.
> - Canal et al., "Active ordinal querying for tuplewise similarity learning," AAAI, 2020.
> - Terada et al., "Local ordinal embedding," International Conference on Machine Learning, 2014.
> - Bower et al., "The landscape of non-convex quadratic feasibility," ICASSP, 2018.
>
> Thank you very much for your detailed feedback which we have worked on to significantly strengthen our work.
>
> If we have answered your questions and have addressed your concerns, we kindly invite you to increase your score. If you still have any more questions, please do let us know and we will do our best to answer them.

---

### Official Review · Reviewer_86F2 · 2025-11-04

**Soundness:** 3
**Presentation:** 2
**Contribution:** 2
**Rating:** 4
**Confidence:** 2

**Summary:**

The paper proposes LORE (Low Rank Ordinal Embedding), an ordinal-embedding framework that jointly learns (i) an embedding that satisfies triplet comparisons and (ii) the intrinsic dimensionality (rank) of the latent perceptual space. The key idea is to regularize the embedding matrix with a nonconvex Schatten-p quasi-norm (with smoothing of the triplet loss), optimized via an iteratively reweighted scheme that is shown to converge to a stationary point. Experiments indicate that LORE achieves comparable triplet accuracy while discovering substantially lower-rank solutions.

**Strengths:**

1 Tackles a long-standing limitation of ordinal embedding, i.e., choosing the dimensionality, by jointly inferring rank and coordinates, rather than grid-searching over dimensions.

2 Uses Schatten-p regularization (p∈(0,1)) to promote low rank, with a softplus-smoothed triplet loss and an iteratively reweighted algorithm; provides a convergence-to-stationary-point guarantee and implementation details.

3 Experiments indicate that LORE achieves comparable triplet accuracy while discovering substantially lower-rank solutions.

**Weaknesses:**

1 The theory ensures convergence to a stationary point, but not global minima or exact rank identification; this is acknowledged as a limitation.

2 The paper argues that LORE uncovers the intrinsic dimensionality without under- or over-estimating it. As stated, this reads as a subjective claim. Please provide stronger evidence to demonstrate that the method does not “mask” latent structure or inflate rank.

3 This paper claims that Künstle et al. (2022) require specifying plausible dimensionalities, risking misspecification and loss of power if the true rank lies outside those bounds. Do you have experiments showing this failure mode and quantifying how often it occurs under realistic sampling/noise?

4 The paper states that training separate embeddings per hypothesized rank (as in Künstle et al., 2022) is computationally prohibitive. Please report the total cost to reach the same triplet accuracy for both methods.

5 Missing direct comparison to Künstle et al. (2022).

6 Literature coverage is dated. The citations lean heavily on pre-2022 works and omit several recent, directly relevant works.

[1] Künstle D E. Machine Learning for Psychophysical Scaling with Ordinal Comparisons[D]. Eberhard Karls Universität Tübingen, 2024.
[2] Huber L S, Künstle D E, Reuter K. Tracing truth through conceptual scaling: Mapping people’s understanding of abstract concepts[J]. 2024.
[3] Sauer Y, Künstle D E, Wichmann F A, et al. An objective measurement approach to quantify the perceived distortions of spectacle lenses[J]. Scientific Reports, 2024, 14(1): 3967.
[4] Huber L S, Künstle D E, Reuter K. Tracing truth through conceptual scaling[J]. Cognition, 2026, 266: 106321.

**Questions:**

1 Can you provide calibration evidence to show that the method neither hides structure nor inflates rank?

2 Do you have experiments where the true rank lies outside the candidate set used by Künstle et al. (2022)? How often does this occur, and what is the performance degradation?

3 For equal target triplet accuracy, what is the cost for LORE vs. training multiple embeddings as in Künstle et al. (2022)?

4 Why is there no quantitative comparison to Künstle et al. (2022)?

5 How does LORE differ conceptually and empirically from recent works?

---

> ### Author Response · Authors · 2025-11-25
> **Official Comment by Authors to Reviewer 86F2**
>
> Thank you very much for the valuable feedback. Below we address your concerns. Please see the revised manuscript (changes in blue, to be removed for camera-ready).
>
> 1. **"Can you provide calibration evidence to show that the method neither hides structure nor inflates rank?"** and "**The paper argues that LORE uncovers the intrinsic dimensionality without under- or over-estimating it. As stated, this reads as a subjective claim. Please provide stronger evidence to demonstrate that the method does not “mask” latent structure or inflate rank.**"
> 	- We agree that empirical evidence is required to show that LORE neither hides structure or inflates rank.
> 		- Real life datasets have no ground truth intrinsic rank so we run simulated experiments that mimic the real data generation process providing ground truth intrinsic rank.
> 		- Section 5.2 and Appendix C show that LORE is robust in obtaining high triplet accuracy (learns the similarity structure) and recovers the true intrinsic rank reliably (does not overparameterize) for the same hyperparameter $\lambda=0.01$ across a variety of dataset conditions (number of triplet queries, number of percepts, dataset noise, and intrinsic rank).
> 		- We add an additional plot to Appendix C varying the intrinsic rank and regularization changes. Section 5.3 and 5.4 meanwhile, compare LORE to other methods in a similarly controlled perceptual experiments where we do have a ground truth intrinsic rank. Again, these results show that LORE neither hides structure nor inflates rank.
> 2. **"Why is there no quantitative comparison to Künstle et al. (2022)?"** and **"The paper states that training separate embeddings per hypothesized rank (as in Künstle et al., 2022) is computationally prohibitive. Please report the total cost to reach the same triplet accuracy for both methods."**
> 	- Thank you for pointing this out! We completely agree that comparisons to (Kunstle et al., 2022), henceforth referred to as Dim-CV, are necessary.
> 		- We reran the simulated perceptual experiment (Section 5.4) and real-life crowdsourced dataset evaluation (Section 5.5) with the Dim-CV method.
> 		- Dim-CV does learn low rank representations, but performs worse than LORE, with a notable cost to triplet accuracy and consistent underfitting relative to OE methods.
> 		- Training multiple embeddings for cross-validation renders Dim-CV nearly two orders of magnitude slower! than other methods, making it impractical for active learning applications.
> 		- Our results indicate that LORE uniquely avoids both underfitting and overparameterizing the perceptual space.
> 3. **"This paper claims that Künstle et al. (2022) require specifying plausible dimensionalities, risking misspecification and loss of power if the true rank lies outside those bounds. Do you have experiments showing this failure mode and quantifying how often it occurs under realistic sampling/noise?"**
> 	- While we provided Dim-CV the full possible intrinsic rank range (1-10) in Sections 5.4 and 5.5, it still failed to recover it consistently.
> 		- In practice do not know what the intrinsic rank will be (without making more assumptions on the rank).
> 		- Even in the most controlled setting for perceptual experiments (same as for other algorithms) of Intrinsic Rank 1-10, Dim-CV cannot recover the true rank.
> 4. **"The paper states that training separate embeddings per hypothesized rank (as in Künstle et al., 2022) is computationally prohibitive. Please report the total cost to reach the same triplet accuracy for both methods."**
> 	- Thank you for this suggestion! We rerun experiments in both Section 5.4 and 5.5 for Dim-CV and find that Dim-CV is almost two orders of magnitude slower than all OE methods with worse accuracy still.
> 		- More CV folds and initializations would take much longer (not feasible to run in rebuttal period).
> 5. **"How does LORE differ conceptually and empirically from recent works?"** and **"Literature coverage is dated. The citations lean heavily on pre-2022 works and omit several recent, directly relevant works."**
> 	- Thank you for pointing out those references! We are familiar with the cited work.
> 		- We (re)include them in the related work (lack of space in original draft)
> 		- The referenced literature consists primarily of applied studies using OE methods for mapping psychophysical/philosophical concepts, rather than offering new methodological contributions. (Huber et al. 2024) and (Sauer et al. 2024)
> 		- (Kunstle 2024) is David E Kunstle's (in my opinion, very well written) PhD thesis that directly cites both Dim-CV (which we cite and we now compare against) and the above mentioned application papers.
> 	- LORE is a methodological contribution that introduces a new OE algorithm that can accurately capture the true perceptual space without underfitting or overparameterizing it like all current OE methods whereas the application papers use existing OE methods.

---

> ### Author Response · Authors · 2025-11-25
> **Official Comment by Authors to 86F2 continued**
>
> **References**
> - Kunstle et al., "Estimating the perceived dimension of psychophysical stimuli using triplet accuracy and hypothesis testing," Journal of Vision, 2022.
> - Kunstle et al., "Machine Learning for Psychophysical Scaling with Ordinal Comparisons," PhD thesis, Eberhard Karls Universität Tübingen, 2024.
> - Huber et al., "Tracing truth through conceptual scaling: Mapping people’s understanding of abstract concepts," 2024.
> - Sauer et al., "An objective measurement approach to quantify the perceived distortions of spectacle lenses," Scientific Reports, vol. 14, no. 1, pp. 3967, 2024.
>
> Thank you very much for your detailed feedback which we have worked on to significantly strengthen our work.
>
> If we have answered your questions and have addressed your concerns, we kindly invite you to increase your score. If you still have any more questions, please do let us know and we will do our best to answer them.

---

### Official Review · Reviewer_sNLD · 2025-11-09

**Soundness:** 3
**Presentation:** 3
**Contribution:** 3
**Rating:** 4
**Confidence:** 3

**Summary:**

The paper claims that all existing OE approaches are based on pre-specified embedding dimensions, which may lead to some problems. Meanwhile, the paper emphasizes the advantages of low intrinsic dimensional embedding--easier to interpret, less computationally intensive, while existing OE approaches based on pre-specified embedding dimensions often result in high-dimensional embeddings.
Based on this, the paper introduces LORE (Low Rank Ordinal Embedding), a novel method for ordinal embedding that jointly learns both the low-dimensional embedding and **its intrinsic dimensionality** from noisy triplet comparisons.
Furthermore, the paper establishes an efficient optimization strategy based on iteratively reweighted minimization and provides a scalable algorithm suitable for large-scale perceptual similarity data.
And the paper validates the effectiveness and efficiency of LOPE through an extensive evaluation

**Strengths:**

1. The explanation of the background and significance of the problem is very clear, and the problem to be solved is very meaningful.
2. LORE is effective in reliably overlooking the intrinsic dimensionality and demonstrates the interpretability of low dimensional representations in semantics, which is helpful for solving problems in psychology, neuroscience, and social science.
3. The theoretical explanation is very rigorous.

**Weaknesses:**

1. More new methods should be compared, and more datasets should be compared, especially considering that SOE and t-STE are both methods from 2014. This may lead to doubts about the performance of LORE.
2. On the accuracy metric, which may be the most important metric, LOPE is not always optimal or even suboptimal.
3. A low rank does not necessarily mean an improvement in method performance, so more explanation is need.
4. For the metric of computational efficiency(time), low dimensional embedding is not the only solution(eg: FORTE vs. LORE), so the advantages of LORE should be further explained.
5. Following 3, if the embeddings of other OE methods are not interpretable, the differences between other methods and LORE should be compared in Figure 5.

**Questions:**

1. If the embeddings of other OE methods are not interpretable, the differences between other methods and LORE should be compared in Figure 5.
2. The factors that affect computational efficiency (time metric) may need to be explained in order for readers to understand why the time differences of methods such as SOE/t-STE can be so significant at the same rank.

---

> ### Author Response · Authors · 2025-11-25
> **Official Comment by Authors to Reviewer sNLD**
>
> Thank you very much for the valuable feedback. Below we address your concerns. Please see the revised manuscript (changes in blue, to be removed for camera-ready).
>
> 1. "**If the embeddings of other OE methods are not interpretable, the differences between other methods and LORE should be compared in Figure 5.**" and "**Following 3, if the embeddings of other OE methods are not interpretable, the differences between other methods and LORE should be compared in Figure 5.**"
> 	- Thank you, we agree that the interpretability of other OE methods should be examined and do so.
> 		- We add plots for the interpretability of all other methods to the Appendix F and see that most methods lack fully interpretable axes. For example:
> 			- SOE splits Sweet to Not Sweet across multiple axes
> 			- FORTE, TSTE and Dim-CV fail to capture all the axes that LORE does
> 			- CKL learns the same axes that LORE does but only LORE is able to directly learn the low rank structure.
> 		-  We add a column to Table 1 indicating the interpretability of the various methods.
> 2. **"The factors that affect computational efficiency (time metric) may need to be explained in order for readers to understand why the time differences of methods such as SOE/t-STE can be so significant at the same rank."**
> 	- Thank you for pointing this out! It is interesting that various OE algorithms can differ so much in processing time even though they all fundamentally solve similar problems.
> 		- We have expanded the background and Section 5.5 to clarify that time taken can result from
> 			- objective function formulation
> 			- optimization method
> 			- dataset characteristics
> 		- As we point out in the background, why one algorithm is faster/slower than another is not straightforward a result that (Vankadara et al, 2023) show in a comprehensive empirical evaluation for ordinal embeddings.
> 3. **"More new methods should be compared, and more datasets should be compared, especially considering that SOE and t-STE are both methods from 2014. This may lead to doubts about the performance of LORE."**
> 	- Thank you very much for the suggestion. We do share the sentiment that our method should be compared to as many recent and high performing OE algorithms and applicable datasets as feasible.
> 		- We have added the Dimensionality Estimation Method (Dim-CV for short) (Kunstle et al. 2022) and reran traces for Section 5.4 and 5.5. LORE outperforms Dim-CV in test triplet accuracy, intrinsic rank estimation and time.
> 		- To the best of our knowledge there is only one (other) recent OE (Suzuki et al. 2019).  We actually attempted to incorporate it for the original draft but its complex implementation involving hyperbolic projections and lack of public code despite no response from multiple emails to the authors resulted in us not being able to include it.
> 		- Datasets: We added results for the Materials similarity dataset (Lagunas et al. 2019) to Section 5.5. LORE achieves top accuracy and matches the "observed" rank via UMAP reported by them. Musicians dataset results are now in the Appendix J.
> 4. **"On the accuracy metric, which may be the most important metric, LOPE is not always optimal or even suboptimal."**
> 	- We do acknowledge on real datasets LORE does not always have the highest test triplet accuracy. However, as we motivate in the introduction, learning the true perceptual representation necessitates having both a high triplet accuracy and learning the true intrinsic rank. This is a property only LORE is able to do.
> 		- Though ground truth intrinsic rank are unknown for real datasets, in simulated experiments we see that LORE is indeed able to recover the intrinsic rank which gives us confidence that LORE will be able to do so in real datasets. (see Section 5.3, 5.4 and Appendix D)
> 		- Other methods fall well short on intrinsic rank estimation whereas LORE is very close on test triplet accuracy or even exceeds in both simulated and real datasets.
> 5. **"A low rank does not necessarily mean an improvement in method performance, so more explanation is need."**
> 	- We agree that a low rank alone does not make a good OE for perceptual experiments.
> 		- As Figure 1 demonstrates (evaluated on real data), there is a tradeoff. Higher rank can capture the similarity relationships better but can overparameterize but a lower rank underfits.
> 		- Recovering the true perceptual space involves balancing both triplet accuracy and intrinsic rank.
> 		- In simulated experiments where intrinsic rank is known from a data generation model, LORE is able to learn both high accuracy and learn the true intrinsic rank giving confidence that it would be similar on real human judgement data.
> 		- We make changes to the introduction and discussion to clarify this further.

---

> ### Author Response · Authors · 2025-11-25
> **Official Comments by Authors to Reviewer sNLD Continued**
>
> 6. **"For the metric of computational efficiency(time), low dimensional embedding is not the only solution(eg: FORTE vs. LORE), so the advantages of LORE should be further explained."**
> 	- Yes, FORTE is faster than LORE in our experiments.
> 		- The speed advantage is due to the kernelized hinge triplet loss and projected gradient descent to optimize in addition to the smaller number of percepts (Jain et al., 2016) (Vankadara et al., 2023).
> 		- However, the speed comes at a cost to accuracy. FORTE's accuracy is consistently lower than SOE, t-STE, CKL and LORE (See Table 2) and Section 5.4.
> 		- FORTE also does not recover the intrinsic rank (see Section 5.3 and 5.4). LORE is consistently quite fast (especially for real datasets) and is the only one able to learn high test triplet accuracy with intrinsic rank recovery.
>
> **References**
> - Kunstle et al., "Estimating the perceived dimension of psychophysical stimuli using triplet accuracy and hypothesis testing," Journal of Vision, 2022.
> - Vankadara et al., "Insights into ordinal embedding algorithms: A systematic evaluation," Journal of Machine Learning Research, 2023.
> - Suzuki et al., "Hyperbolic ordinal embedding," Asian Conference on Machine Learning, 2019.
> - Jain et al., "Finite sample prediction and recovery bounds for ordinal embedding," Advances in Neural Information Processing Systems, 2016.
> - Lagunas et al., "A similarity measure for material appearance," arXiv preprint, 2019.
>
> Thank you very much for your detailed feedback which we have worked on to significantly strengthen our work.
>
> If we have answered your questions and have addressed your concerns, we kindly invite you to increase your score. If you still have any more questions, please do let us know and we will do our best to answer them.

---

### Author Response · Authors · 2025-11-26
**Official Comment by Authors**

We sincerely thank the reviewers for the detailed feedback. We are glad that the reviewers highlight some of the strengths of our work.

1. **Addressing a fundamental limitation of Ordinal Embeddings:** *LORE successfully addresses a long standing challenge of choosing the dimensionality to embed the representation for ordinal embeddings.* (sNLD, kWHX, inQi, 86F2)
2. **Theoretical Rigor:** *Unlike many empirical approaches, LORE provides a rigorous convergence guarantee for its optimization objective and training algorithm* (sNLD, kWHX, inQi, 86F2)
3. **Interpretability:** *Representations learned by LORE are highly interpretable and semantically meaningful* (sNLD, inQi, kWHX)
4. **Parsimony:** *LORE is able to successfully learn embeddings with high triplet accuracy but with substantially lower rank* (86F2, inQi, kWHX)

We have resolved the following primary concerns shared by multiple reviewers in addition to specific individual questions.
1. **We now also compare to Kunstle et al 2022 / new methods** - We evaluate LORE against the method from Kunstle et al 2022 and find that LORE outperforms on test triplet accuracy, intrinsic rank and time taken. (sNLD, 86F2)
2. **Validation on diverse human judgement datasets** - We add results on the Materials similarity dataset. We now comprehensively evaluate LORE on **four disparate human perceptual similarity datasets**. We find that LORE outperforms all methods on test triplet accuracy while consistently recovering lower-rank representations, demonstrating strong generalization to real-world data. (sNLD)
3. **Stability and Convergence of Initialization**. We provide new experiments analyzing the effect of initialization on convergence and runtime. We demonstrate that **initialization does not affect the final objective** obtained by LORE across a variety of settings, confirming the method's stability. (kWHX, inQi)
4. **Robustness to hyperparameters** - We clarify that LORE is not sensitive to hyperparameter tuning. The method relies on a single hyperparameter, and a fixed setting ($\lambda=0.01$) successfully recovers high accuracy and intrinsic rank across tasks. Our extensive controlled experiments that comprehensively vary characteristics of ordinal data show that LORE is not sensitive to hyperparameter choice. This ensures that LORE is plug and play with no hyperparameter tuning for practitioners in the perceptual sciences. (86F2, inQi, kWHX)
5. **Comparative Interpretability** - We have added interpretability plots for competing OE methods. Our results show that **only LORE and CKL** are capable of learning fully interpretable ordinal embeddings. (sNLD)
6. **Reproducibility:** We have updated the manuscript with comprehensive experimental setup details to ensure full reproducibility. (inQi, kWHX)

If we have successfully addressed your questions, we invite you to kindly increase your score. If not, please do let us know and we will do our best in answering them.

---

### Meta-Review · Area_Chair_SqDa · 2026-01-08

**Summary:**

The paper introduces a method to construct an ordinal embedding (embedding of data given ordinal triplet comparisons) while putting a penalty on the embedding dimensionality, effectively resulting in dimensionality estimation. The method builds up on SOE (2014) and adds a regularizer to it, pushing singular values of the embedding towards zero (in a lasso-like way).

The reviewers liked the paper but complained about insufficient comparisons, insufficient datasets, and insufficient simulated experiments with known ground truth.

**Reviewer Concerns:**

The authors did a very extensive rebuttal, providing detailed responses to all reviews, and uploading a revised manuscript. The revised version contains new experiments and new comparisons with more recent methods and on more datasets. This does largely address many of the concerns.

I like the revised paper. My personal two biggest concerns are: (1) The effect of p (Schatten norm parameter) is not really investigated. The authors make a big deal about using the Schatten norm with p<1 instead of nuclear norm, which corresponds to p=1. But they simply fix p=0.5 and never show any results with p=1. This makes me wonder if the nuclear norm would work as well. It would be good to see some comparisons with varying p.

(2) Comparison to Dim-CV in Figure 4 (added in the revision) shows that Dim-CV fails to recover the true rank. Why is this the case? As far as I understood, Dim-CV simply constructs an embedding for different dimensionalities and choose the lowest one with high accuracy. So why does that fail here? In fact, what you show is the "measured rank" (which you did not define) which seems to be fractional, whereas Dim-CV should yield an integer "estimated rank". Can you show the estimated rank of Dim-CV in this experiment? If it does not grow with the intrinsic rank, then why not? This failure mode of Dim-CV should be analyzed in more detail, because this is the most relevant comparison for LORE: instead of adding penalty, fit embedding in different dimensionalities and choose the lowest sufficient one.

Minor: Equation in line 241 lacks lambda. The effective rank (line 322) is never defined.

**Reviewer Scores:**

The original scores were 4/4/4/6. I feel the extensive rebuttal does address various raised issues via additional experiments, so some reviewers would have raised their scores. I think the final scores could have been 4/6/6/6. I think it is still borderline, but I personally liked the paper, so am recommending acceptance. If accepted, please do address the issues I listed above.

---

### Decision · Program_Chairs · 2026-01-26

Accept (Poster)